# WHERE DID THIS SENTENCE COME FROM? TRACING PROVENANCE IN LLM REASONING DISTILLATION

**Kaiyuan Liu[1], Shaotian Yan[2], Rui Miao[3], Bing Wang[4], Chen Shen[2†], Jun Zhang[5], Jieping Ye[2†]**

[1]College of Computer Science and Technology, Zhejiang University
[2]Alibaba Cloud Computing
[3]School of Artificial Intelligence, Jilin University
[4]College of Computer Science and Technology, Jilin University
[5]Department of Mathematics, University of Michigan
12421281@zju.edu.cn

## ABSTRACT

Reasoning distillation, a cost-effective approach for enhancing student model performance, has attracted increasing attention. It typically leverages a large teacher model to generate reasoning paths, which are then used to fine-tune a student model so that it mimics the teacher's behavior in training contexts. However, previous approaches have lacked a detailed analysis of the origins of the distilled model's capabilities. It remains unclear whether the student can maintain consistent behaviors with the teacher in novel test-time contexts, or whether it regresses to its original output patterns, raising concerns about the generalization of distillation models. To analyse this question, we introduce a cross-model Reasoning Distillation Provenance Tracing framework. For each action (e.g., a sentence) produced by the distilled model, we obtain the predictive probabilities assigned by the teacher, the original student, and the distilled model under the same context. By comparing these probabilities, we classify each action into four categories: (i) teacher-originated actions, (ii) student-originated actions, (iii) pre-existing actions in both models not enhanced by distillation, and (iv) pre-existing actions boosted through distillation. By systematically disentangling the provenance of each action, we experimentally demonstrate that, in test-time contexts, the distilled model can indeed generate teacher-originated actions, which correlate with and plausibly explain observed performance on distilled model. Building on this analysis, we further propose a teacher-guided data selection method. Unlike prior approach that rely on heuristics (e.g., selecting data most aligned with the student's original distribution), our method directly compares teacher–student divergences on the training data, providing a principled selection criterion. We validate the effectiveness of our approach across multiple representative teacher models (Deepseek, QwQ, GPT-OSS-120B) and diverse student models (Qwen2.5-7B-Instruct, Qwen3-4B-Base, Qwen3-8B-Base, Qwen3-4B-Instruct-2507). The results highlight the utility of our provenance-tracing framework and underscore its promise for reasoning distillation. We hope to share Reasoning Distillation Provenance Tracing, along with our insights into reasoning distillation, with the community.

## 1 INTRODUCTION

The rapid development of reinforcement learning (RL) techniques (Schulman et al., 2017; Shao et al., 2024; Ahmadian et al., 2024) and resulting large-scale reasoning models (DeepSeek-AI, 2025; OpenAI, 2025; Team, 2025b) has accelerated the growth of distillation researches, especially reasoning distillation (Zhao et al., 2025; Guha et al., 2025). Early work on reasoning distillation focused on creating high-quality open-source datasets (Zhao et al., 2025; Guha et al., 2025; NVIDIA, 2025), and recent studies have focused on curating and filtering distillation samples to improve efficiency and performance (Zhang et al., 2025; Li et al., 2025). However, these approaches primarily focus on model performance and fail to provide an explanatory analysis of the sources of the model's output

---

† Corresponding author

in test-time contexts. As a result, it remains unclear whether the student model has successfully inherited the knowledge and reasoning logic from the teacher model, raising concerns about the generalization of distillation models (Hinton et al., 2015b).

Concretely, as illustrated in Figure 1, reasoning distillation involves two main steps. First, specific contexts (e.g., high-quality questions, or high-quality questions augmented with partially generated answers) are provided to the teacher model, which then produces the next action via sampling. Second, these context–action pairs are used to train the student model, encouraging it to reproduce the teacher's actions under the same contexts. At test time, however, the distilled student faces new contexts. It remains unclear whether the student will continue to follow the teacher's behavior instead of falling back on its original output distribution.

To analyse the above problem, we propose a cross-model Reasoning Distillation Provenance Tracing framework. Concretely, we evaluate three open-source distilled models: DeepSeek-Distill-Qwen-7B (DeepSeek-AI, 2025), DeepSeek-R1-0528-Qwen3-8B (DeepSeek-AI, 2025), and LIMO-v2 model (Ye et al., 2025). To analyze the patterns in distilled models' outputs under test contexts, we collect multiple responses from them on GPQA-D (Rein et al., 2024) and AIME24. Each response is re-input into the distilled model, the corresponding teacher model, and the original student model. This procedure allows us to obtain, under the same test scenario, the probability assigned by each model to the next action produced by the distilled model. By comparing these probabilities, we can naturally trace the provenance of every action. For example, if the teacher model assigns significantly higher probability to an action than the original student model does, we can attribute that action to teacher-originated actions, since the distilled model's ability to produce it mainly derives from teacher model's knowledge transferred during distillation. For further details about other action types, see Section 3.

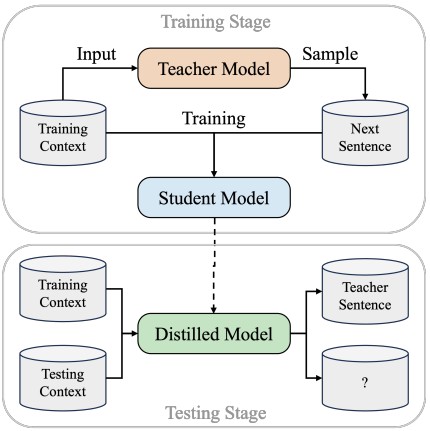

Figure 1: Motivation. In reasoning distillation, the student model learns in the training stage to produce actions consistent with the teacher model within the training context. However, at test time, it remains unclear whether the distilled model will continue to output actions aligned with the teacher model instead of degrading to the outputs of the student model, raising concerns about the generalization of distillation models (Hinton et al., 2015b).

To this end, we observe that distilled models can reproduce teacher-originated actions in new test contexts. Moreover, these actions are correlated with correct responses on the test set, which helps explain the generalization gains achieved through distillation. This observation further motivates our training design: we hypothesize that when the training data contains a higher proportion of teacher-originated actions, the distilled model attains better final performance. To validate this hypothesis, we propose a teacher-guided data selection strategy that compares the probabilities assigned by the teacher and student models on the training data and prioritizes examples that contain more teacher-originated actions. We then evaluate its effectiveness across multiple settings, including representative teacher models from different families (DeepSeek-R1-671B (DeepSeek-AI, 2025), QwQ-32B (Team, 2025b), GPT-OSS-120B (OpenAI, 2025)) and various student models (Qwen2.5-7B-Instruct, Qwen3-4B-Base, Qwen3-8B-Base, Qwen3-4B-Instruct-2507).

Our contributions can be summarised as follows: (1) We propose Reasoning Distillation Provenance Tracing, a systematic method to disentangle the origins of each action, through fine-grained categorization into teacher-originated, student-originated, shared, and boosted actions. This offers a principled approach for analyzing whether distillation genuinely transfers reasoning ability or merely reinforces pre-existing patterns. (2) Through analysis on reasoning benchmarks, we show that distilled models can generate teacher-originated actions even in unseen test scenarios. These actions are correlated with correctness, offering a quantitative explanation for why reasoning distillation improves generalization. (3) Building on the provenance analysis, we introduce a teacher-guided data selection strategy that prioritizes training samples rich in teacher-originated actions. Unlike heuristic method, our approach leverages explicit teacher–student divergence as a selection criterion. Experiments across diverse teacher–student pairs demonstrating performance gains in our settings.

## 2 RELATED WORK

### 2.1 REASONING DISTILLATION

Distilling the reasoning abilities of large reasoning models has been an important problem since their emergence (Kim & Rush, 2016; Ho et al., 2023; Ko et al., 2024; 2025). DeepSeek (DeepSeek-AI, 2025) pioneered this line of work by showing that supervised fine-tuning on the outputs of a reasoning teacher, which is also the approach we focus on in this paper, can dramatically enhance the reasoning abilities of smaller models. Numerous subsequent projects (e.g., HuggingFace OpenR1 (Hugging Face, 2025), OpenThoughts (Guha et al., 2025), a-m-team (Zhao et al., 2025), NVIDIA AceReason (Liu et al., 2025b), Alibaba OmniThought (Guha et al., 2025), LIMO (Ye et al., 2025), Tencent DeepMath (He et al., 2025)) have devoted substantial effort to constructing large-scale corpora of challenging reasoning problems paired with teacher responses, using rigorous quality filtering, correctness checks, and diversity-aware curation. Most recently, GRAPE (Zhang et al., 2025) preferentially selects examples whose likelihoods best match the student's current distribution, thereby steering training toward data that is already well aligned with the student. **Rather than focusing solely on artificially designed rules and heuristic rules, we aim to quantify the sources of a distilled model's capabilities and introduce a data selection criterion that focuses on sentences whose probabilities indicate stronger teacher-originated behavior. This provenance-aware criterion complements prior student-only selection and provides an explicit cross-model signal for reasoning transfer.**

### 2.2 MODEL AUDITING

Another closely related area is model auditing, a growing line of work that studies (Song & Shmatikov, 2019; Carlini et al., 2022; Dekoninck et al., 2024) auditing generative models to understand what data they memorize and to attribute outputs back to underlying data sources. **In contrast, our work targets model-level provenance in a distillation setting: rather than asking whether specific data are memorized, we aim to trace which upstream models are the sources of a given output, shifting the focus from data membership to the lineage of the models themselves.**

Due to page limitations, we provide further discussions on related work in the Appendix A.6.

## 3 REASONING DISTILLATION PROVENANCE TRACING

In this section, we first provide necessary notations in Subsection 3.1 and introduce Reasoning Distillation Provenance Tracing in Subsection 3.2. We then apply Reasoning Distillation Provenance Tracing to three widely used open-source models (Deepseek-Distill-Qwen-7B, DeepSeek-R1-0528-Qwen3-8B and LIMO-v2 model) and present the results of the analysis in Subsection 3.3.

### 3.1 NOTATIONS

Specifically, let $M_T$ denote the teacher model, $M_S$ the student model, and $\{Q^i_{train}\}$ a set of high-quality training questions, where $Q^i_{train}$ denotes the $i$-th question. Reasoning distillation first samples responses from $M_T$ on $\{Q^i_{train}\}$, yielding the training set $\mathcal{D}_{train} = \{(Q^i_{train}, \tau^i)\}$, where $\tau_i = \{a_{(i,j)}|j = 1, \cdots, L_i\}$ denotes the trajectory for the $i$-th question, consisting of $L_i$ actions generated by $M_T$. The student model $M_S$ is then trained by minimizing the cross-entropy loss between its predicted next-token-actions and the teacher's actions in trajectory $\tau_i$ under the same context (e.g., input $(Q^i_{train}, a_{(i,1)}, a_{(i,2)})$, output $a_{(i,3)}$). This ensures that distilled model $M_D$ produces reasoning trajectories mimicking to those of $M_T$ when presented with the same context.

### 3.2 METHOD

As stated in the Section 1, our goal is to analyze whether the distilled model $M_D$ can still produce actions similar to those of the teacher model $M_T$ in new contexts. The most direct approach is to input the same context into both models ($M_T$ and $M_D$) and compare their actions. Yet this approach faces two challenges. First, reasoning outputs are typically long, and even when segmented step by step, the number of sentences remains large. Iteratively truncating at each sentence boundary and re-input both models to generate new actions is prohibitively expensive. Second, it is difficult to accurately evaluate the similarity between the newly generated actions of the two models.

To address these issues, we shift perspective and instead sample exclusively from the distilled model $M_D$ on the test set $\{Q^i_{test}\}$. We first obtain the trajectory on test set $\mathcal{D}_{test} = \{(Q^i_{test}, \tau^i)\}$. Subsequently, we feed $\tau^i$ back into the three models ($M_T$, $M_S$, and $M_D$) and analyse each component action $a_{(i,j)}$. For each action, since token-level comparison is sometimes difficult due to possible

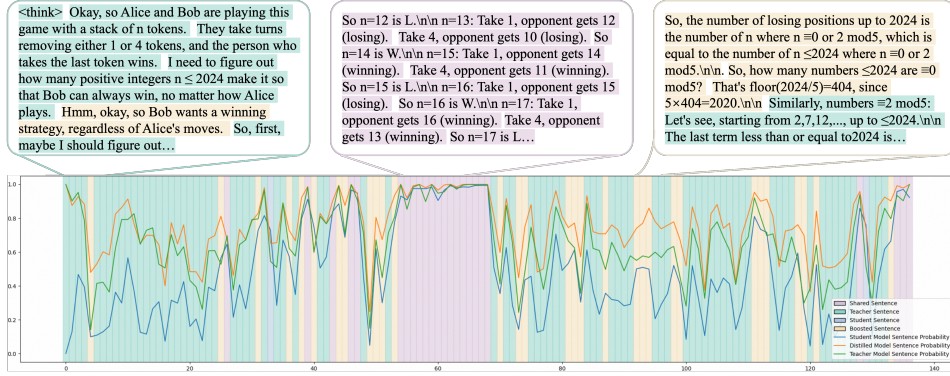

Figure 2: An illustration of analysis using Reasoning Distillation Provenance Tracing. The horizontal axis denotes the action position (i.e., the sentence order), while the vertical axis shows the probability assigned by different models (indicated by colored curves in the foreground) in producing that action under the same context. The background colors indicate the action types. Some example of different action types are also shown at the top for illustration, where the blank spaces indicate the segmentation boundaries of the response.

vocabulary mismatches between $M_T$ and $M_D$. We use a coarser alternative, comparing at the sentence level. In this way, we define the probability of producing $a_{(i,j)}$ as the geometric mean of per-token probabilities $p_{(i,j)} = exp(mean(log(p_k)))$, where $p_k$ denote the probability of $k$-th token contained with the sentence $a_{(i,j)}$.

For each $a_{(i,j)}$, we can obtain three output probabilities under the same context: $p^T_{(i,j)}$ from $M_T$, $p^S_{(i,j)}$ from $M_S$, and $p^D_{(i,j)}$ from $M_D$. As shown in Figure 2, the blue line corresponds to $p^S_{(i,j)}$, the orange line to $p^D_{(i,j)}$, and the green line to $p^T_{(i,j)}$.

Overall, since the trajectory $\tau^i$ is sampled from the distilled model $M_D$, the orange curve ($p^D_{(i,j)}$) tends to be the largest on average. Furthermore, four distinct patterns can be observed. Specifically:

**Pre-existing actions in both models not enhanced by distillation** (hereafter referred to as Shared Sentence): This includes actions such as those between the 54th and 68th steps, where the output probabilities of all models are nearly identical. These actions are originally present in both the teacher and the student models, and distillation does not further increase their probabilities.

**Pre-existing actions boosted through distillation** (hereafter referred to as Boosted Sentence): Similar to the first type, $p^T_{(i,j)}$ and $p^S_{(i,j)}$ remain close, but $p^D_{(i,j)}$ differs significantly (and is typically higher in practice, since trajectories are sampled from $M_D$). These actions also exist in both the teacher and student models prior to distillation, but their probabilities are amplified through training with distilled data.

**Student-originated actions** (hereafter referred to as Student Sentence) and **teacher-originated actions** (hereafter referred to as Teacher Sentence): When there is a large discrepancy between $p^S_{(i,j)}$ and $p^T_{(i,j)}$, the distilled model $M_D$ still outputs the action, suggesting the action is more consistent with the model assigning higher likelihood. Note that a Teacher Sentence does not imply that the action is entirely absent from the student model, but rather that it is primarily originated from the teacher. The same applies to a Student Sentence.

### 3.3 ANALYSIS ON OPEN-SOURCE MODELS

In this subsection, we apply Reasoning Distillation Provenance Tracing to DeepSeek-Distill-Qwen-7B (Ds-7B), DeepSeek-R1-0528-Qwen3-8B (Ds-8B) and LIMO-v2 model to analyze the source of each sentence produced by the distilled models in new testing context. Specifically, we sample the open-source models on two commonly used reasoning benchmarks: AIME24 and GPQA-D. AIME24 uses 16 completions per question, and GPQA-D uses 8.

For each completed response, we segment the output according to the following rules: (1) Special tokens (e.g., <think>) are treated as individual actions, as they carry specific se-

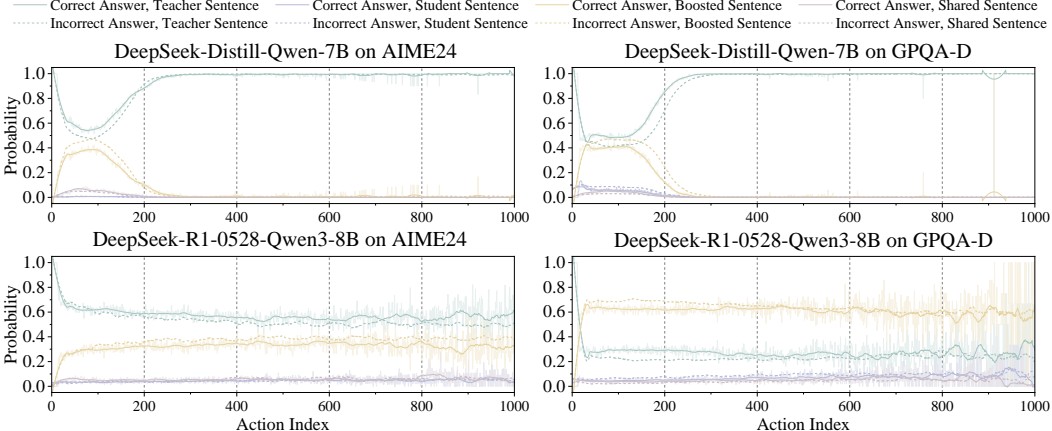

Figure 3: Analysis results on open-source models. We apply Reasoning Distillation Provenance Tracing to Deepseek-Distill-Qwen-7B and DeepSeek-R1-0528-Qwen3-8B, analyzing the probability of producing different actions at each action position on two reasoning benchmarks (AIME24 and GPQA-D). The x-axis denotes the action index, and the y-axis shows the probability of producing that action.

mantics and represent important behavior. (2) For the remaining text, we split sentences using the pattern: punctuation + optional whitespace + uppercase letter, corresponding to: `re.compile(r'([.?!}\]])([\s\n]+)([A-Z])')`. This segmentation procedure covers most common English sentence structures.

Then, we feed the sampled trajectories back into the corresponding teacher and student models to obtain the output probability for each token, and consequently (by taken geometric average of token probability), the probability for each action ($p_{(i,j)}^S$, $p_{(i,j)}^D$ and $p_{(i,j)}^T$). We then define $\Delta_{SD} = p_{(i,j)}^S - p_{(i,j)}^D$, $\Delta_{TD} = p_{(i,j)}^T - p_{(i,j)}^D$ and $\Delta_{TS} = p_{(i,j)}^T - p_{(i,j)}^S$. Using two thresholds, $\alpha$ and $\beta$, we can perform the tracing provenance:

$$
\begin{cases}
\text{Shared Sentence,} & \textbf{if}(|\Delta_{SD}| \leq \alpha \,\wedge\, |\Delta_{TD}| \leq \alpha \,\wedge\, |\Delta_{TS}| \leq \alpha), \\
\text{Teacher Sentence,} & \textbf{else if}(\Delta_{TS} > \beta), \\
\text{Student Sentence,} & \textbf{else if}(-\Delta_{TS} > \beta), \\
\text{Boosted Sentence,} & \textbf{else if}(|\Delta_{TS}| < \beta).
\end{cases}
$$

where $p_{(i,j)}^D$, $p_{(i,j)}^S$ and $p_{(i,j)}^T$ denote the probabilities assigned to action $a_{(i,j)}$ by the distilled model $M_D$, the student model $M_S$, and the teacher model $M_T$, respectively. We follow the evaluation order: Shared → Teacher → Student → Boosted. The role of $\alpha$ is to filter out relatively small probability differences (such as those between sentences 55 and 65 in Figure 2) to prevent them from influencing the analysis. $\beta$ helps to more clearly differentiate between various action types, and its value can be determined adaptively. Due to page limitations, the selection of $\alpha$ and $\beta$ is detailed in Appendix A.2.

Based on this classification scheme, we compute, for each action position, the proportion of each action type and interpret this proportion as the probability of outputting that action type at the given position. For example, for all answers at the third action position (i.e., the third sentence), we calculate the proportion of actions belonging to Teacher Sentence and treat it as the probability that the model outputs Teacher Sentence at that position. **It is worth noting that the number of actions varies across different answers; therefore, we focus primarily on the earlier action positions, where sufficient data is available to yield reliable statistics.** In addition, we compute the average number of tokens for all actions at each position, and, at 4k-token intervals, mark in the action positions corresponding to the average number of tokens required to reach that position. The results are shown in Figure 3 and Figure 4.

We observe three phenomena (see Appendix A.3 for more analyses) that help explain the observed benefits of reasoning distillation in novel test-time settings:

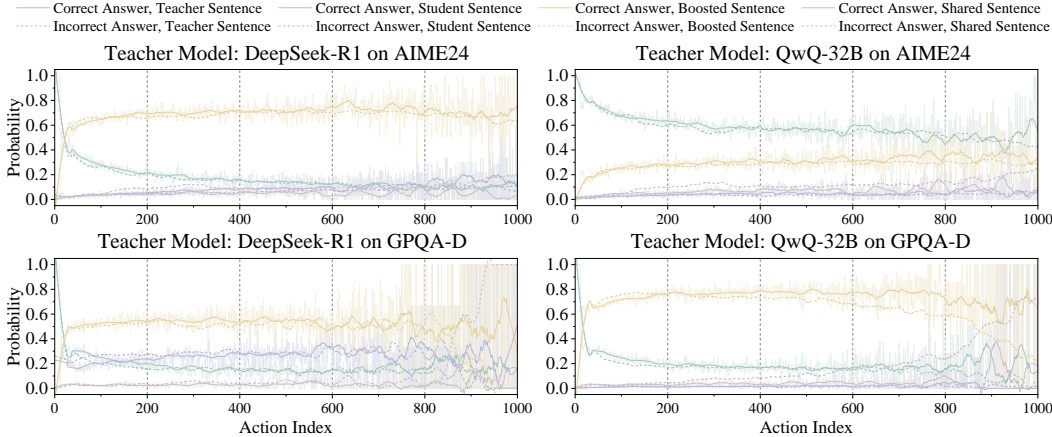

Figure 4: Analysis results on LIMO-v2 model (Ye et al., 2025). We apply Reasoning Distillation Provenance Tracing to LIMO-v2 model, analyzing the probability of producing different actions at each action position on two reasoning benchmarks (AIME24 and GPQA-D). Since LIMO-v2 model uses multiple teacher models, we tried different teacher models on all samples. The x-axis denotes the action index, and the y-axis shows the probability of producing that action.

**(1) Higher Teacher Sentence probability in the early inference stage.**

As shown in Figure 3 and Figure 4, the light-green line (—) exhibits larger values in the early action index, meaning that the probability of outputting a Teacher Sentence is relatively high in the early stages of inference. We attribute this to two factors. First, the student model lacks the ability to properly generate the token `<think>`, and subsequent steps may be influenced by the first action. However, this effect is limited, as evidenced by the rapid increase in the probability of Boosted Sentence (i.e., the light-yellow line (—) rises sharply in the early phase.). Figure 2 and Figure 11 further illustrate this point, showing that other actions typically emerge after only a few steps. Second, we observe that many early actions focus on analyzing the input and planning subsequent steps. We hypothesize that forcing such behavior early on may be a unique pattern of the teacher model, which explains why Teacher Sentences are more likely to appear at the beginning of inference.

**(2) Student-internal patterns are often activated but not uniformly beneficial during reasoning distillation.**

In LIMO-v2 paper, the authors found that if a model already contains sufficient reasoning knowledge, reasoning distillation can activate it with minimal data scale and yield strong performance. In this work, we find that reasoning distillation not only activates reasoning knowledge, but also triggers other latent patterns embedded within the Boosted Sentence. As shown in Figure 3 and Figure 4, we observe that the mean of the pointwise sum of the light-green (—) and light-yellow (—) lines exceeds 0.7, indicating that at each position the predicted action is likely to be one of these two action types. This means that, in distilled models, the majority of output action types are either Teacher Sentences or Boosted Sentences. More Teacher Sentences originate from the student model learning directly from the teacher model. And more Boosted Sentences suggests that reasoning distillation consistently activates student-internal patterns.

To analyze how reasoning distillation helps in novel contexts, we further conduct a quantitative analysis of action-type probabilities in correct versus incorrect responses. As shown in Figure 4, on LIMO-v2 model, Boosted Sentences emerge in later stages with consistently high probability in correct responses across different teacher models (i.e., in later stages, the light-yellow solid line (—) remains above the light-yellow dashed line (- - -)). In the early stages, correct responses are more likely to rely on Teacher Sentences (i.e., the light-green solid line (—) stays above the light-green dashed line (- - -)). However, Figure 3 also reveals the opposite trend in smaller models: Boosted Sentences consistently appear with higher probability in incorrect responses. This suggests that not all student-internal patterns are worth activating. The authors of LIMO-v2 paper also find that small models (e.g., 7B) trained with the same method fail to achieve good performance. Taken together with our findings, one concludes that reasoning distillation often activates internal patterns in student model, but not all of these activations are beneficial.

**(3) Teacher sentences are highly correlated with performance gains.**
As shown in Figure 3, across different models (DeepSeek-R1-0528-Qwen3-8B and DeepSeek-Distill-Qwen-7B) and test sets (AIME24 and GPQA-D), Teacher Sentence tends to be assigned higher probabilities to correct answers (i.e., the light-green solid line (—) stays above the light-green dashed line (- - -)). This result is also intuitive: since the teacher model achieves higher performance on the test sets, the student model benefits from producing outputs more aligned with the teacher, thereby increasing its likelihood of answering correctly.

It remains an open question whether Teacher Sentence is equally beneficial for larger models, and we believe further experimentation is needed. Although Figure 4 clearly shows that Teacher Sentence assigns higher probabilities to correct answers in the early stages (while probabilities for correct and incorrect answers converge in later stages), the evidence is limited. LIMO-v2 dataset contains only 800 samples, designed primarily to activate student-internal patterns and validate conclusions in the original paper. Thus, it is unclear whether, at larger training scales, Teacher Sentence would still yield significantly higher probabilities for correct answers in the mid-to-late stages.Due to resource constraints, this work focuses on models of 8B parameters or fewer. We hope to validate our findings on larger models in future work.

**In summary, our Reasoning Distillation Provenance Tracing shows that in new test settings, distilled models not only produce sentences originating from the teacher, but these Teacher Sentence are especially prominent in early reasoning phases and associated with answer correctness. At the same time, reasoning distillation can also activate patterns already latent within the student (Boosted Sentences), whose contributions vary by model size and reasoning phases, and are not uniformly beneficial.**

## 4 BEYOND EXPLANATION, GUIDING TRAINING IN REVERSE: TEACHER-GUIDED DATA SELECTION

Building on the analysis in Section 3, a natural training insight is: **if we can increase the proportion of Teacher Sentence in the training data and explicitly leverage teacher–student divergences to select examples, we may front-load and amplify the observed generalization benefits.** Therefore, in this section, we first propose a teacher-guided data selection method in Section 4.1. Then we validate its effectiveness across multiple teacher–student pairs and benchmarks in Section 4.2.

### 4.1 TEACHER-GUIDED DATA SELECTION

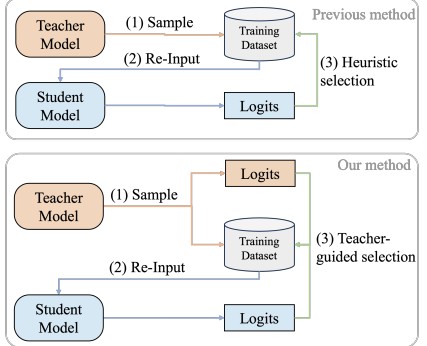

Figure 5: Comparison between teacher-guided data selection and the previous method (Zhang et al., 2025).

To validate our hypothesis, the first question we must address is: how can Reasoning Distillation Provenance Tracing be applied prior to training? At this stage, we only have access to the teacher model and the student model. Nevertheless, even with only these two models, it is still possible to reliably distinguish between Teacher Sentences and Student Sentences. Specifically:

$$\begin{cases} \text{Common Sentence} & \textbf{if}(|\Delta_{TS}| \leq \beta), \\ \text{Teacher Sentence} & \textbf{else if}(\Delta_{TS} > \beta), \\ \text{Student Sentence} & \textbf{else if}(-\Delta_{TS} > \beta). \end{cases}$$

where Common Sentence subsumes the previously defined Shared Sentence and Boosted Sentence. Through this approach, we are able to effectively filter training data to include more Teacher Sentence before training. Specifically, for each question with multiple responses from the same teacher model, we count the number of Teacher Sentences in each response and prioritize for training the response with the largest count.

As shown in Figure 5, the previous method (Zhang et al., 2025) feeds the training data into the student model before training, computes heuristics metrics based on the logits of student model, and selects samples that more aligned with the student model's original distribution. In contrast, **our proposed method provides a clearer objective for data selection: prioritizing samples where the teacher and student models differ the most**.

Moreover, our method introduces only an acceptable additional cost in practice. (1) When one wishes to sample responses, similar to the previous method (Zhang et al., 2025), our approach

incurs no extra overhead, since logits can be obtained directly during sampling. (2) When one wishes to leverage existing open-source distillation datasets for filtering, some additional cost is required. However, it is important to note that re-feeding the generated sequences into the model to extract logits only requires a single forward pass, which is significantly faster than token-by-token generation. Given sufficient GPU memory, we consider the additional time cost to be acceptable.

## 4.2 EXPERIMENTS

We aim to evaluate the effectiveness of teacher-guided data selection across diverse training settings. To this end, we conduct experiments with three distinct teacher models (Deepseek-R1-671B, QwQ-32B, and GPT-OSS-120B) and four student models: Qwen3-4B-Base, Qwen3-8B-Base, Qwen2.5-7B-Instruct, and Qwen3-4B-Instruct-2507. We use two high-quality reasoning datasets, AceReason-1.1-SFT (Liu et al., 2025b) and OpenThought3-1.2M (Guha et al., 2025). Each question in these datasets is paired with multiple candidate answers sampled from the teacher model. Further details are provided in Appendix A.5. For each question, we then select a single response using one of three strategies: (i) random selection, which performs similarly to vanilla reasoning distillation ("Vanilla"); (ii) GRAPE (Zhang et al., 2025) ("GRAPE"); and (iii) our method ("Ours"). All strategies yield training sets of identical size.

For evaluation, we select four widely adopted benchmarks known for their challenging reasoning demands: AIME24, AIME25, MATH500 (Lightman et al., 2023), and OlympiadBench (He et al., 2024). We report the accuracy results.

### 4.2.1 MAIN EXPERIMENTS

As shown in Tables 1, our method achieves the best performance. On average, it improves results by 1.7%–2.5%.

Table 1: Comparison with different data selection strategy. Each experimental setting is denoted as Teacher Model + Student Model + Data Source.

| | AIME24 | AIME25 | MATH500 | OlympiadBench | Average |
|---|---|---|---|---|---|
| *Deepseek-R1 + Qwen3-4B-Base + AceReason-1.1-SFT* | | | | | |
| Vanilla | 44.4 | 33.3 | **91.2** | 55.9 | 56.2 |
| GRAPE (Zhang et al., 2025) | 43.0 | 34.1 | 88.6 | 54.8 | 55.1 |
| Ours | **49.3**$_{+4.9}$ | **37.9**$_{+4.6}$ | 90.8$_{-0.4}$ | **56.6**$_{+0.7}$ | **58.7**$_{+2.5}$ |
| *Deepseek-R1 + Qwen3-8B-Base + AceReason-1.1-SFT* | | | | | |
| Vanilla | 55.1 | 39.9 | 91.2 | 57.5 | 60.9 |
| GRAPE (Zhang et al., 2025) | 54.2 | 38.1 | 91.8 | 58.4 | 60.6 |
| Ours | **57.3**$_{+2.2}$ | **41.5**$_{+1.6}$ | **92.8**$_{+1.6}$ | **58.7**$_{+1.2}$ | **62.6**$_{+1.7}$ |
| *QwQ-32B + Qwen2.5-7B-Instruct + OpenThought3-1.2M* | | | | | |
| Vanilla | 43.5 | 35.6 | 89.8 | 55.1 | 56.0 |
| GRAPE (Zhang et al., 2025) | 47.5 | 34.6 | **91.4** | 54.8 | 57.1 |
| Ours | **48.1**$_{+4.6}$ | **36.3**$_{+0.7}$ | 90.0$_{+0.2}$ | **56.3**$_{+1.2}$ | **57.7**$_{+1.7}$ |
| *GPT-OSS-120B + Qwen3-4B-Instruct-2507 + AceReason-1.1-SFT* | | | | | |
| Vanilla | 75.9 | 62.5 | 93.6 | 64.0 | 74.0 |
| GRAPE (Zhang et al., 2025) | 76.8 | 66.5 | 92.6 | 62.5 | 74.6 |
| Ours | **77.9**$_{+2.0}$ | **68.3**$_{+5.8}$ | **94.6**$_{+1.0}$ | **64.9**$_{+0.9}$ | **76.4**$_{+2.4}$ |

Table 2: Influence of different selection metrics.

| | AIME24 | AIME25 | MATH500 | OlympiadBench | Average |
|---|---|---|---|---|---|
| Maximize Absolute Count | 49.3 | 37.9 | 90.8 | 56.6 | 58.7 |
| Longest | 48.1 | 37.5 | 90.0 | 55.9 | 57.9 |
| Relative Proportion | 46.9 | 35.0 | 88.8 | 55.1 | 56.4 |
| Vanilla | 44.4 | 33.3 | 91.2 | 55.9 | 56.2 |
| Minimize Absolute Count | 42.9 | 35.2 | 87.8 | 54.2 | 55.0 |

### 4.2.2 ABLATION EXPERIMENTS

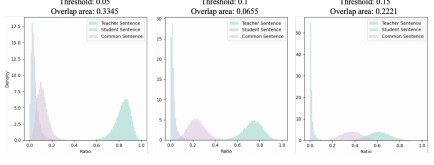

Figure 6: Illustration of $\beta$ selection for the first training setting.

**(1) How to Leverage Teacher Sentences for Data Selection?** We generally aim to include more Teacher Sentences in training data, where "more" can be interpreted in two ways: by the absolute count of Teacher Sentences within a response ("Maximize Absolute Count") or by their relative proportion ("Relative Proportion"). Figure 3 shows that, across most action positions in correct responses, Teacher Sentences receive higher output proba-

bilities from the distilled model. This observation suggests two principles for effective data selection: (i) choose responses that contain as many reasoning actions as possible ("Longest") so that the training signal can influence the maximum number of action positions, and (ii) increase the absolute number of those actions that correspond to Teacher Sentences ("Maximize Absolute Count") to raise the model's probability of producing them at those positions.

To compare these metrics, we conduct an ablation study using Deepseek-R1 as the teacher, Qwen3-4B-Base as the student, and AceReason-1.1-SFT as the data source. The results in Table 2 show that **Maximize Absolute Count yields the best performance, whereas Minimize Absolute Count performs the worst, indirectly corroborating our hypothesis.**

Table 3: Impact of different $\beta$ values on accuracy.

|  | AIME24 | AIME25 | MATH500 | OlympiadBench | Average |
|---|---|---|---|---|---|
| $\beta = 0.05$ | 47.9 | 37.1 | 89.8 | 57.0 | 58.0 |
| $\beta = 0.1$ | 49.3 | 37.9 | 90.8 | 56.6 | 58.7 |
| $\beta = 0.15$ | 46.3 | 37.5 | 90.2 | 56.3 | 57.6 |

**(2) How to determine $\beta$?** In Section 3, we describe how to set $\beta$ at test time. Here, we assess whether the same procedure is effective on the training data. We conduct validation experiments using DeepSeek-R1 as the teacher model, Qwen3-4B-Base as the student model, and AceReason-1.1-SFT as the training data source. As shown in Figure 6, before training we observe that $\beta = 0.1$ produces the cleanest partition, with minimal overlap across action types. As shown in Table 3, the optimal $\beta$ determined before training achieved the best results in the ablation experiment. Moreover, other near-optimal $\beta$ also outperform the Vanilla baseline, indicating that performance is not highly sensitive to the choice of $\beta$. **Together, the post-training results, along with the pre-training analysis, consistently support our data-driven $\beta$-selection strategy as a principled, training-free, and effective approach.** For further discussion, see Appendix A.2.

**(3) Effects in different domain.** To evaluate the effectiveness of our method in different domain, we used GPT-OSS-120B to sample 10k questions from OpenScienceReasoning_2 (NVIDIA Corporation, 2024), generating nine candidate responses per question. We then selected a single response for training using different selection strategies. Results are reported in Table 4. We find that our method is effective in the scientific domain and, compared to GRAPE, exhibits stronger generalization in math domain. Analysis on the test set further indicates that the Teacher Sentence in science shares commonalities with that in mathematics, which we hypothesize helps explain why training on scientific data can improve performance on the math test set. Additional analyses are provided in Appendix A.3.3.

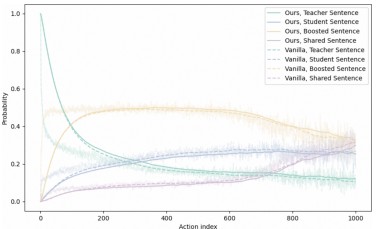

Figure 7: Analysis results of the first training setting (Deepseek-R1 + Qwen3-4B-Base + AceReason-1.1-SFT) on AIME24.

Table 4: Comparison with different data selection strategy on scientific domain.

|  | GPQA-D | AIME24 | AIME25 | MATH500 | OlympiadBench |
|---|---|---|---|---|---|
| *GPT-OSS-120B + Qwen3-4B-Base + OpenScienceReasoning_2* | | | | | |
| Vanilla | 39.9 | 24.2 | 24.6 | 85.8 | 44.0 |
| GRAPE (Zhang et al., 2025) | 40.9 | 24.6 | 22.1 | 84.2 | 44.3 |
| Ours | **42.9**$_{+3.0}$ | **25.2**$_{+1.0}$ | **24.8**$_{+0.2}$ | **87.4**$_{+1.6}$ | **48.0**$_{+4.0}$ |

**(4) Does the distilled model increase the probability of outputting the Teacher Sentence?** We apply the Reasoning Distillation Provenance Tracing to our trained models. The results are shown in Figure 7. Compared with the Vanilla baseline, within the range where estimates are statistically reliable (e.g., the first 800 action indices), our method increases the probability of outputting the Teacher Sentence and correspondingly decreases that of the Student Sentence.

## 5 CONCLUSION

In this work, we address a fundamental yet under-explored question in reasoning distillation: does the distilled model truly inherit the teacher's pattern? To answer this, we introduce Reasoning Distillation Provenance Tracing, observe phenomena and quantify the evidences that help explain the observed benefits of reasoning distillation. Building on these insights, we propose a teacher-guided data-selection strategy and demonstrate its effectiveness on multiple settings. We hope our provenance-tracing framework will inspire future research on cross-model behavior analysis, domain-aware data selection, and more reliable distillation protocols for complex reasoning tasks.

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

# A APPENDIX

## A.1 FUTURE WORK

Future work will pursue three directions: (1) when more logits information is available, combine sequential probability matching methods to improve our data selection; (2) expand beyond the small-model regime studied here, imposed by resource constraints, to evaluate the effectiveness of our data-selection strategy on larger models; and (3) move past the single-teacher setting to investigate principled approaches to data selection when training with ensembles of diverse teacher models.

## A.2 HOW TO DETERMINE $\alpha$ AND $\beta$

### A.2.1 HOW TO DETERMINE $\alpha$

For $\alpha$, it is used to filter out relatively small probability differences that would otherwise affect the analysis. To obtain a relatively objective threshold, we asked 10 annotators to judge the magnitude of probability differences. For each annotation, the annotators were shown an output-probability line chart similar to that in Figure 2. Given all the sentences in the response, we asked them to identify those whose probability differences could be regarded as relatively negligible, such as sentences 55–65 in Figure 2. After we explained the definitions of the four sentence types and the goal of our analysis, the annotators manually selected the sentences that appeared more likely to be Shared Sentences, and we then derived the final rounded value of $\alpha = 0.1$ from their judgments.

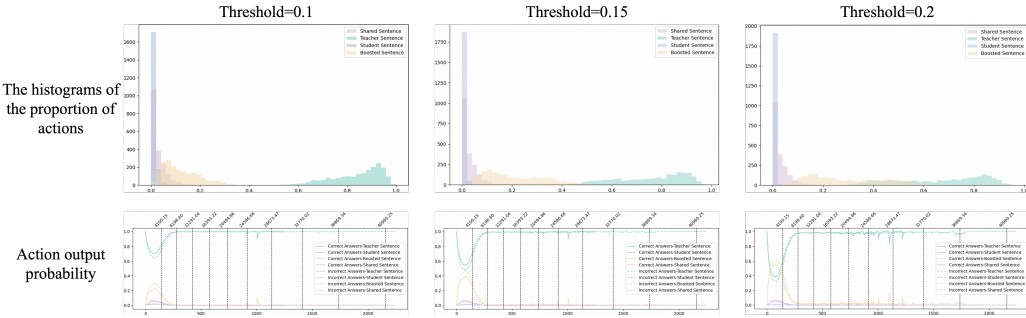

Figure 8: Illustration of $\beta$ selection for Deepseek-Distill-Qwen-7B on AIME24.

### A.2.2 HOW TO DETERMINE $\beta$

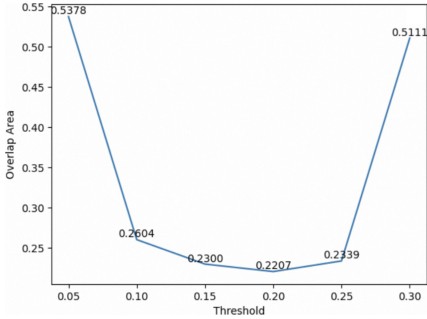

Figure 9: Illustration of $\beta$ selection process.

The parameter $\beta$ serves primarily to facilitate a clearer separation of the relative proportions of different action types, enabling more effective provenance analysis. Specifically, for each trajectory, we compute the proportion of sentence attributed to each action type (e.g., in trajectory 1, 60% of sentences are labeled as Teacher Sentence; in trajectory 2, 70%). We then construct histograms showing the distribution of these proportions across all samples for each action type (e.g., plotting the distribution of Teacher Sentence proportions using values such as [0.6, 0.7, ...]).

Intuitively, we choose $\beta$ to maximize the separation between the distributions of different action types, so that their overlap is minimized, particularly for the action types that dominate in frequency. As shown in Figure 8, for instance, when $\beta = 0.15$, the histogram reveals that most samples exhibit a clear predominance of Teacher Sentence, providing a clean characterization of the model's output behavior. In contrast, when $\beta = 0.2$, the distributions of Teacher Sentence and Boosted Sentence exhibit significant overlap, indicating that samples within the overlapping region are sensitive to minor fluctuations and may be ambiguously classified. A similar problem also exists with $\beta = 0.1$.

---

**Algorithm 1** Search for Optimal $\beta$

---

**Require:** teacher model $M_T$, student model $M_S$, batch of data $\mathcal{D}$
**Ensure:** optimal threshold $\beta^\star$
 1: Feed $\mathcal{D}$ into $M_T$ and $M_S$ to obtain sentence-level output probabilities.
 2: Following Section 4.2.2, partition sentences and get sentence-level probabilities into: Common
    Sentence set $\mathcal{C}$, Teacher Sentence set $\mathcal{T}$, Student Sentence set $\mathcal{S}$.
 3: best overlap: $O^\star \leftarrow +\infty$
 4: best $\beta$: $\beta^\star \leftarrow$ None
 5: **for** $\beta \in \{0.05, 0.10, \ldots, 1.0\}$ **do**
 6:     Compute the histogram overlap between $\mathcal{S}$ and $\mathcal{C}$ under threshold $\beta$, denote it as $O_1$.
 7:     Compute the histogram overlap between $\mathcal{C}$ and $\mathcal{T}$ under threshold $\beta$, denote it as $O_2$.
 8:     Current overlap $O \leftarrow O_1 + O_2$
 9:     **if** $O < O^\star$ **then**
10:         $O^\star \leftarrow O$
11:         $\beta^\star \leftarrow \beta$
12:     **end if**
13:     **if** $\mathrm{mean}(\mathcal{C}) > \mathrm{mean}(\mathcal{T})$ **then**
14:         **break**
15:     **end if**
16: **end for**
17: **return** $\beta^\star$

---

Therefore, for Deepseek-Distill-Qwen-7B on AIME24, we set $\beta = 0.15$. For the experiments in Section 3.3, we evaluate all values of $\beta$ in the range [0.05, 0.2] with a step size of 0.05. The final selected thresholds are: $\beta = 0.1$ for Deepseek-Distill-Qwen-7B on GPQA-D, $\beta = 0.1$ for DeepSeek-R1-0528-Qwen3-8B on AIME24, $\beta = 0.15$ for DeepSeek-R1-0528-Qwen3-8B on GPQA-D, $\beta = 0.1$ for LIMO-v2 model on AIME24 when using QwQ-32B as teacher model, $\beta = 0.2$ for LIMO-v2 model on GPQA-D when using QwQ-32B as teacher model, $\beta = 0.15$ for LIMO-v2 model on AIME24 when using Deepseek-R1 as teacher model, and $\beta = 0.1$ for LIMO-v2 model on AIME24 when using Deepseek-R1 as teacher model.

In fact, threshold selection is also an adaptive procedure rather than a manually specified parameter that must be tuned via repeated training runs, which makes it fundamentally different from hyperparameter selection in neural networks. Taking the training-time pipeline as an example, we perform the search using Algorithm 1. For the four training configurations in Table 1, the chosen $\beta$ values are 0.1, 0.1, 0.15, and 0.2, respectively. In addition, we show how this algorithm selects the value 0.2 for Table 1, Setting 4, as illustrated in Figure 9.

### A.2.3 SENSITIVITY ANALYSIS

We also illustrate threshold selection on first training setting (DeepSeek-R1 + Qwen3-4B-Base + AceReason-1.1-SFT) in Figure 6. The corresponding quantitative results are reported in Table 3. As can be seen, the threshold identified as optimal before training also yields the best performance after training. The post-training metrics in Table 3 and the pre-training visualizations in Figure 6 demonstrate that the proposed method is purely data-driven, requires no additional training, and is practically applicable. Near-optimal $\beta$ also outperform the Vanilla baseline, indicating that performance is not highly sensitive to the choice of $\beta$.

Additionally, we examine the sensitivity of our analysis to the choice of $\beta$. As shown in Figure 8, the relative trends and distinctions in action-type output probability between correct and incorrect samples remain consistent across different $\beta$ values, suggesting that our main conclusions are robust to the $\beta$ setting.

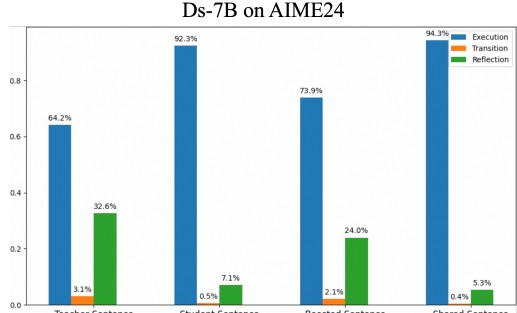 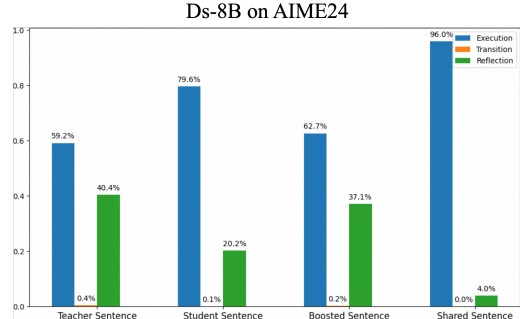

Figure 10: Differences across action types. Using SEAL's behavioral categorization, we classify all actions and report the proportion of behavior types within each action type.

## A.3 MORE ANALYSIS RESULTS

### A.3.1 ALTHOUGH DISTILLED MODELS GENERALLY EXHIBIT SUBSTANTIALLY STRONGER LONG-CONTEXT GENERATION CAPABILITIES, THE SOURCES OF THESE IMPROVEMENTS ARE NOT UNIFORM ACROSS MODELS.

As shown in Figure 3, within DeepSeek-Distill-Qwen-7B, once the sequence length exceeds approximately 4K tokens, the generated outputs are quickly transformed into Teacher Sentence. In contrast, this behavior is not observed in DeepSeek-R1-0528-Qwen3-8B or LIMO-v2 model. We attribute this discrepancy to differences in the underlying student model: DeepSeek-Distill-Qwen-7B is distilled from Qwen2.5-Math-7B (Yang et al., 2024), which has an effective context length of approximately 4K tokens. Beyond this limit, the model's outputs predominantly reflect patterns inherited from the teacher model. By comparison, DeepSeek-R1-0528-Qwen3-8B and LIMO-v2 model are based on Qwen3-8B-Base (Team, 2025a) and Qwen2.5-32B-Instruct, respectively. These models support substantially longer effective context lengths and thus avoid this limitation.

### A.3.2 WHAT BEHAVIORS ARE INCLUDED IN DIFFERENT ACTION TYPES?

We categorize each action using SEAL (Chen et al., 2025), which defines three types of behaviors: execution, reflection, and transition. Execution refers to steps that directly advance problem solving, reflection denotes verification, checking, or questioning of the existing reasoning process, and transition represents an intentional change in the current reasoning direction or strategy. Special tokens are excluded from the statistics. The results are shown in Figure 10, from which we make the following three observations.

(1) Both student models exhibit a certain degree of reflection and transition, and reasoning distillation further activates and strengthens these behaviors. Specifically, Student Sentences already contain reflection and transition behaviors, while Boosted Sentences display more of these behaviors than Student Sentences and Shared Sentences after distillation. Although there remains a gap between RL and reasoning distillation, this observation is similar with prior work (Wang et al., 2025; Gandhi et al., 2025), which suggests that reflection and transition abilities are already latent in student models and that RL/Reasoning distillation as a post-training method serves to activate them. (2) The vast majority of Shared Sentences are Execution. (3) The student model of Deepseek-R1-0528-Qwen3-8B (Ds-8B) shows stronger reflective ability than the student model of Deepseek-Distill-Qwen-7B (Ds-7B), as evidenced by the higher proportion of reflection behaviors in Student Sentences.

### A.3.3 TEACHER SENTENCES ACROSS DOMAINS

As illustrated in Figure 11, we further examine the characteristics of Teacher Sentences across domains. In mathematics, Teacher Sentences primarily consist of explicit mathematical operations, followed by checks of these operations and validations of the solution logic. In science, they more often involve inspections of reasoning chains and targeted recall of relevant knowledge. Despite these differences, we observe common patterns: when reasoning stalls, Teacher Sentences prompt the recall of key facts and encourage reflective adjustments to the reasoning process. We hypothe-

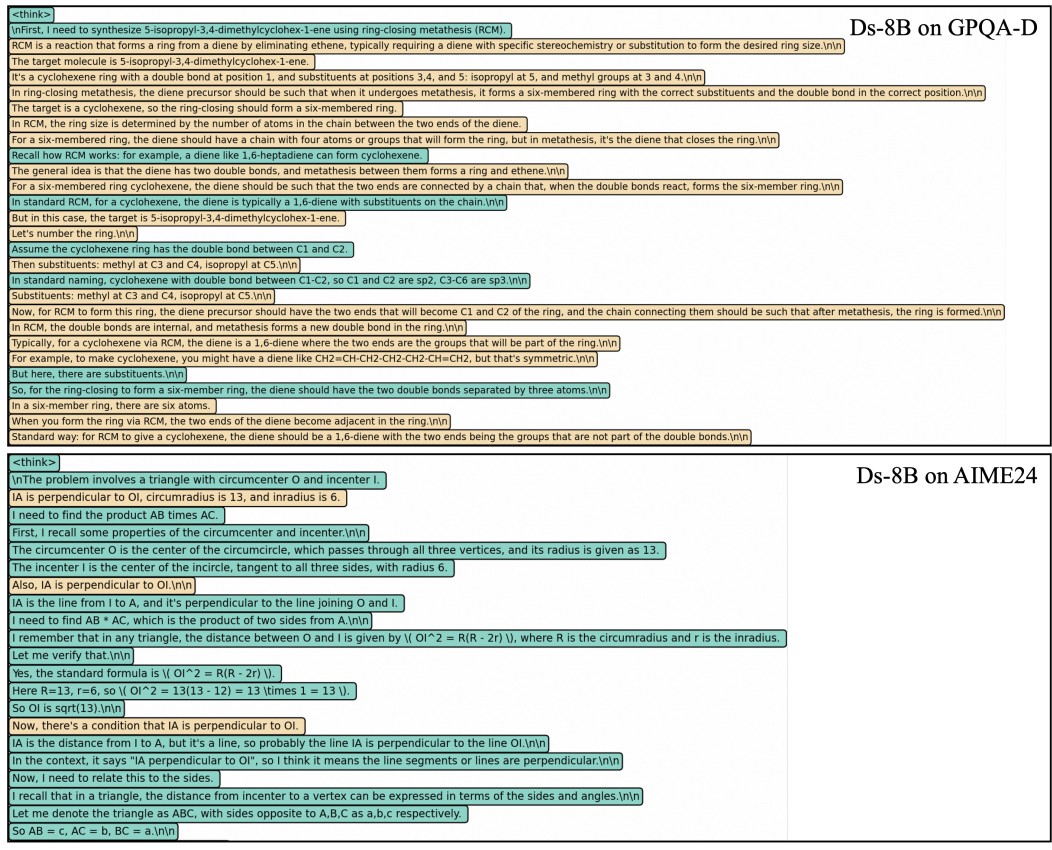

Figure 11: Comparison of Teacher Sentences across domains. Teacher Sentences are highlighted in light green.

Table 5: Training config.

|  | Qwen3-4B-Base | Qwen3-8B-Base | Qwen3-4B-Instruct-2507 | Qwen2.5-7B-Instruct |
|---|---|---|---|---|
| learning_rate | 5e-5 | 5e-5 | 5e-5 | 8.0e-5 |
| cutoff_len | 32k | 64k | 64k | 16k |
| epoch | 6 | 6 | 6 | 5 |
| batchsize | 32 | 32 | 32 | 512 |
| lr_scheduler_type | cosine_with_min_lr | cosine_with_min_lr | cosine_with_min_lr | cosine |
| min_lr | 1e-5 | 1e-5 | 1e-5 | 0 |
| warmup_ratio | 0.1 | 0.1 | 0.1 | 0.1 |

size that these shared behaviors explain why training on scientific-domain data containing a higher proportion of Teacher Sentences can also benefit performance on mathematics test sets.

## A.4 MORE DISCUSSIONS

### A.4.1 A NEW PERSPECTIVE ON UNDERSTANDING MODELS

Most existing approaches Lee et al. (2025); Liu et al. (2025a) to understanding model outputs operate in a human-centric semantic space. They typically seek to decompose intermediate activations or outputs into human-interpretable units, such as natural language concepts, symbolic structures, or predefined semantic categories. While this line of work has yielded valuable insights, it implicitly assumes that model behavior is best understood by mapping it onto human-designed semantic structures.

We agree that semantic/structure-level analyses are important. But we believe that decomposing the outputs of a reasoning model into human-interpretable semantic units is not the only way to

understand model behavior. Instead, our method offers a novel and complementary perspective: we construct a decomposition that is naturally induced by the model itself and directly ties its outputs to performance on the test set. This model-centric view allows us to analyze how different components of the model's output contribute to its empirical performance, without requiring a predefined set of human-interpretable semantic units.

### A.4.2 ALIGNMENT OF OUTPUT DISTRIBUTIONS

**In the distillation setting, we define better reasoning as closer alignment with the teacher model's output distribution. The goal of reasoning distillation is therefore twofold: (i) the student model should learn the teacher model's output distribution, and (ii) this alignment should yield better performance on the test set.** In addition to the existing analysis of the second point (in main text), we further examine the first point. To examine (i), we construct three sets of sentences: the teacher model's outputs on the training set, the teacher model's outputs on the test set, and the distilled student model's outputs on the test set. For each set, we feed every response into both the teacher model and the distilled student model, compute the difference between the output probabilities for each sentence, and then aggregate these differences to obtain the empirical distribution histogram of probability discrepancies.

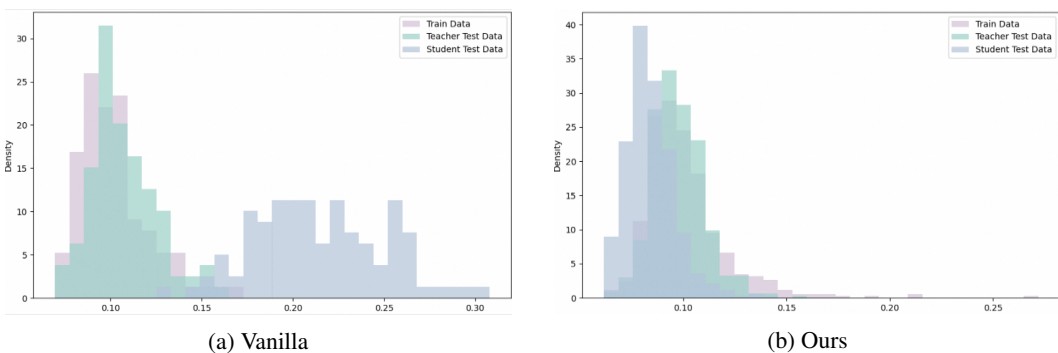

(a) Vanilla        (b) Ours

Figure 12: Distribution of output probability differences. "Train Data", "Teacher Test Data", and "Student Test Data" represent the outputs of the teacher model on the training set, the outputs of the teacher model on the test set, and the outputs of the distilled student model on the test set, respectively.

The results, shown in the Figure 12, compare two different data-filtering strategies. Under both strategies, the probability discrepancy is small on the training data (the teacher's outputs on the training set), which is expected because the training data are well fitted. The discrepancy on the teacher's outputs on the test set is slightly larger, indicating that, although the queries in the training and test sets differ, the student model still largely follows the teacher model's output distribution when evaluated on the teacher's context. Remarkably, when evaluated on the distilled model's own context on the test set, our method achieves an even smaller probability discrepancy than on the training data. This strongly suggests a high degree of alignment with the teacher's output distribution. It is important to note that the teacher distribution is not directly observable. We only approximate it indirectly via the probabilities assigned to sentences under different contexts. The fact, that the smallest probability discrepancy occurs on the distilled model's own outputs on the test set, indicates that our method allows the student model, in its own generation context, to closely align with the teacher model's output distribution.

### A.4.3 CAN THE PROVENANCE METHODOLOGY BE USED TO IDENTIFY WHICH TEACHER MODEL DISTILLED THE REASONING CAPABILITY?

In short, our method is in principle capable of providing evidence about which teacher model contributed to a given reasoning capability. To demonstrate this ability, we consider two different scenarios and validate it in each of them.

(1) The student model is given, but its teacher model are unknown.

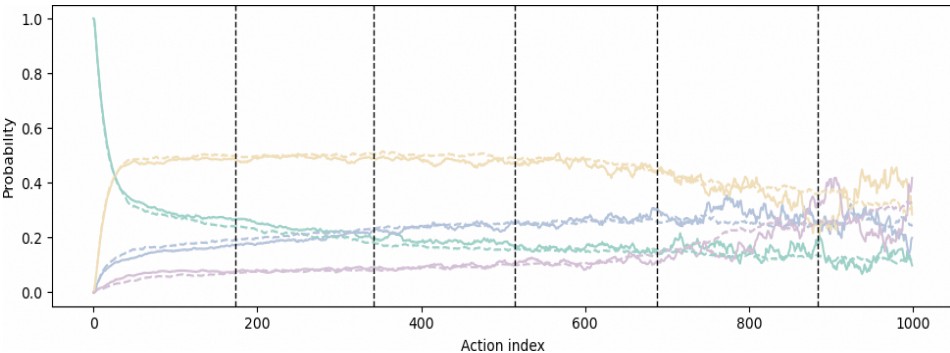

Figure 13: Analysis results on **original teacher**. We apply Reasoning Distillation Provenance Tracing to Deepseek-Distill-Qwen-7B and DeepSeek-R1-0528-Qwen3-8B, analyzing the probability of producing different actions at each action position on two reasoning benchmarks (AIME24 and GPQA-D). The x-axis denotes the action index, and the y-axis shows the probability of producing that action.

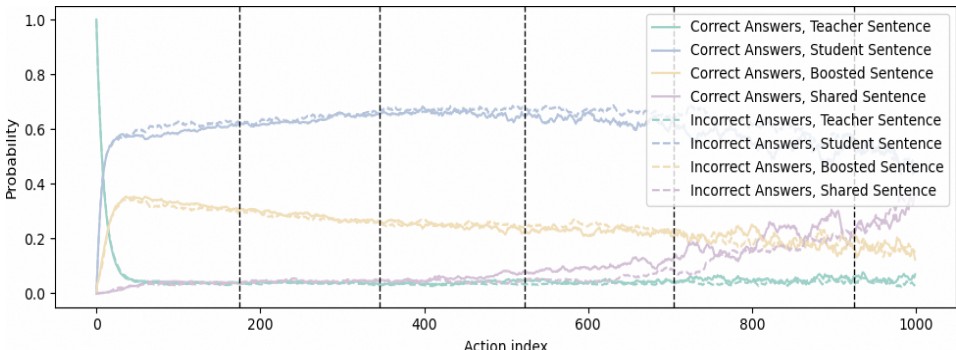

Figure 14: Analysis results on **spurious teacher**. We apply Reasoning Distillation Provenance Tracing to Deepseek-Distill-Qwen-7B and DeepSeek-R1-0528-Qwen3-8B, analyzing the probability of producing different actions at each action position on two reasoning benchmarks (AIME24 and GPQA-D). The x-axis denotes the action index, and the y-axis shows the probability of producing that action.

We employ the distilled model from setting 1 in Table 1, which was in fact distilled from DeepSeek-R1. We then apply our analysis method to two candidate teacher models (i.e., DeepSeek-R1 and gpt-oss-120b) to determine which teacher was actually used for distillation.

The analysis results for DeepSeek-R1 are shown in Figure 13, while results for gpt-oss-120b are presented in the Figure 14. We observe that, when a spurious teacher model is used, the output probability of the Teacher Sentence is the lowest, while the output probability of the Student Sentence is relatively high. This indicates that our analysis method can, to some extent, identify the true teacher model behind a distilled student model.

(2) The student model is known to be distilled from multiple teacher models.

From a methodological perspective, our provenance analysis can partially identify the extent to which the teacher model contributes to the student model's reasoning capability. For example, in our analysis of the LIMO-v2 model, we observe that, across all teacher models, the Teacher Sentence tends to be assigned a higher output probability on correct answers than on incorrect ones. However, different teacher models exert heterogeneous influences on the student model. Concretely, over the first 400 steps (those with relatively stronger statistical significance), we compute the cumulative difference between the output probability of the Teacher Sentence on correct versus incorrect answers, and obtain the following sums: gpqa-R1 (9.41), gpqa-QwQ (7.66), aime-R1 (7.09), aime-QwQ (7.88).

These results suggest that, although the exact proportions of R1- and QwQ-generated data in the LIMO-v2 training set are unknown, the data produced by these two teacher models have heterogeneous effects across domains. We tentatively hypothesize that, on GPQA, the reasoning capability

distilled from R1 is particularly beneficial for scientific questions. The result is consistent with the fact that DeepSeek-R1 substantially outperforms QwQ on GPQA while performing comparably to QwQ on AIME. In future work, we plan to further investigate how to identify which teacher model has contributed the distilled reasoning capability to the student model on different domains.

## A.5 Training Details

For the OpenThought3-1.2M dataset, we used only the mathematical problems and randomly sampled 50k questions. For each question, the official dataset provides 16 responses generated by QwQ; from these, we randomly selected 8 responses to form our initial training set.

For the AceReason-1.1-SFT dataset, we first randomly sampled 50k mathematical questions. For each question, the official dataset provides multiple responses generated by R1. We then additionally generated 1–5 responses per question using GPT-OSS-120B (configured with `Reasoning: high`). Together, the R1 and GPT-OSS-120B responses served as our initial training set.

The training configuration is provided in Table 5.

## A.6 Related Work

### A.6.1 LLM Distillation

**Knowledge Distillation**. Knowledge distillation was first introduced by (Hinton et al., 2015a) as a technique for transferring the dark knowledge of a teacher model to a lightweight student model via soft labels, thereby enabling substantial model compression while preserving most of the original accuracy. Subsequent works (SE, 2014; Zagoruyko & Komodakis, 2016) has extended this framework from multiple perspectives, for example by distilling feature layers or intermediate representations and by introducing relational or structural distillation, so that the student not only mimics the output distribution but also aligns the structural properties of the hidden representation space. Later works (Ko et al., 2024; 2025) further improve alignment by approximating sequence-level KL divergence, leading to more effective distillation. In practice, knowledge distillation (Gou et al., 2021) has been widely applied to tasks such as image classification, object detection, and natural language processing, and has become one of the mainstream approaches for model compression and acceleration.

**Reasoning Distillation**. Distilling the reasoning abilities of large language models has been an important problem since their emergence (Kim & Rush, 2016; Ho et al., 2023; Ko et al., 2024; 2025). Before the advent of large-scale reasoning models such as O1 (Jaech et al., 2024), traditional reasoning distillation methods primarily transferred capabilities by aligning intermediate features or output probabilities between teacher and student models (Kim & Rush, 2016; Ho et al., 2023; Ko et al., 2024; 2025). To teach reasoning more explicitly, prior work (Hsieh et al., 2023; Kim & Rush, 2016; Ho et al., 2023; Ko et al., 2024; 2025) constructs responses that include detailed reasoning traces and trains the student on these signals, thereby strengthening its mastery of reasoning. In the era of large-scale reasoning models such as R1 (DeepSeek-AI, 2025) and QwQ (Team, 2025b), which naturally exhibit chain-of-thought reasoning and achieve strong performance, distilling their capabilities into smaller models has become an effective and practical path toward improved efficiency. DeepSeek (DeepSeek-AI, 2025) pioneered this line of work by showing that supervised fine-tuning on the outputs of a reasoning teacher, which is also the approach we focus on in this paper, can dramatically enhance the reasoning abilities of smaller models. Numerous subsequent projects (e.g., OpenR1 (Hugging Face, 2025), OpenThoughts (Guha et al., 2025), a-m-team (Zhao et al., 2025), NVIDIA AceReason (Liu et al., 2025b), OmniThought (Guha et al., 2025), LIMO (Ye et al., 2025), DeepMath (He et al., 2025)) have devoted substantial effort to constructing and refining large-scale corpora of challenging reasoning problems paired with teacher responses, using rigorous quality filtering, correctness checks, and diversity-aware curation. Most recently, GRAPE (Zhang et al., 2025) scores candidate responses with the student model and preferentially selects examples whose likelihoods best match the student's current distribution, thereby steering training toward data that is already well aligned with the student. **Rather than focusing solely on artificially designed rules and heuristic rules, we view reasoning distillation as a capability-transfer problem from teacher to student. We aim to quantify the sources of a distilled model's capabilities: given a context, which actions in a trajectory from distilled model are more likely to originate from the teacher's behavior rather than the student's existing tendencies? Building on this perspec-**

**tive, we introduce a data selection criterion that jointly compares teacher–student output distributions and focuses on sentences whose probabilities indicate stronger teacher-originated behavior. This provenance-aware criterion complements prior student-only selection in the following way: it provides an explicit cross-model signal for reasoning transfer.** In Section 4, we show that provenance-aware selection outperforms student-only alignment in our settings.

### A.6.2 MODEL AUDITING

Another closely related area is model auditing, a growing line of work that studies (Song & Shmatikov, 2019; Carlini et al., 2022; Dekoninck et al., 2024) auditing generative models to understand what data they memorize and to attribute outputs back to underlying data sources. For example, prior work (Song & Shmatikov, 2019) shows that rare tokens in the training data tend to be memorized by text generation models, and uses shadow models together with an audit classifier (e.g., an SVM on token-rank features) to distinguish whether a user's data was included in training. Separately, subsequent work (Carlini et al., 2022) formalizes extractability as the ability of a model, given a prefix, to greedily regenerate the exact suffix from the training set, and systematically studies how repetition and sequence length affect the fraction of such extractable sequences. **In contrast, our work targets model-level provenance in a distillation setting: rather than asking whether specific data are memorized, we aim to trace which upstream models are the sources of a given output, shifting the focus from data membership to the lineage of the models themselves.**

### A.7 LLM USAGE

We used Qwen3 for polishing, followed by manual refinement.

