# OpenReview forum: "Where Did This Sentence Come From? Tracing Provenance in LLM Reasoning Distillation"
_ICLR.cc/2026/Conference — ICLR 2026 Poster_

### Official Review · Reviewer_E2xa · 2025-10-22

**Soundness:** 2
**Presentation:** 2
**Contribution:** 3
**Rating:** 4
**Confidence:** 3

**Summary:**

This submission studies reasoning distillation, where a smaller student model is fine-tuned on reasoning paths generated by a larger teacher model. It first proposes a method called Reasoning Distillation Provenance Tracing to categorize each reasoning step ("action") produced by a distilled model, based on the probabilities assigned to it by the distilled model, teacher model, and student model before distillation. The aim is to determine whether a reasoning step is more likely to originate from the teacher model, from the student before distillation, or be "boosted" by distillation even though the teacher does not assign it high probability. The authors present a study using Reasoning Distillation Provenance Tracing on open-source distilled models and reasoning datasets. The second part of the submission builds on the first by selecting examples for distillation where the probability of the example's actions being teacher-generated is higher. Experiments show that this data selection strategy outperforms random selection and a baseline called GRAPE in terms of the distilled model's performance on reasoning benchmarks.

**Strengths:**

- This work aims to go beyond performance on reasoning benchmarks to understand the effects that distillation has on a granular, step-by-step level. The proposed categorization of actions appears simple and reasonable.
- The work also aims to close the loop by using the categorization to select more effective examples for distillation, with encouraging results.
- An effort is made to experiment on multiple student models, teacher models, distillation datasets, and evaluation datasets.

**Weaknesses:**

1. The paper claims that distilled models reproduce teacher-originated actions in "entirely new contexts," and the newness of test contexts is emphasized throughout. However, this newness is not substantiated. Moreover, newness as a binary notion misses the fact that similarity to training contexts lies on a spectrum. I think it would be better to quantify somehow the dissimilarity of test contexts from training contexts, at both the sample and dataset levels, and analyze results in terms of this dissimilarity.
1. Figures are unclear, to the point of obscuring evidence for main claims of the paper (see especially the first point below).
    - Figures 3 and 4: The solid and dashed lines are hard to distinguish. This makes it hard to be convinced of finding (3) in Section 3.3 in particular, i.e., the correlation between Teacher Sentences and correctness. Can this correlation be quantified so that the evidence is not just visual comparison of two curves? It is also hard to understand finding (2) in Section 3.3 from the current figures.
    - Legends, tick labels, etc. are too small: This matters especially in Figures 3 and 4 because the line colors and line types are not described in the text.
    - Figure 6: The x-axis is not labelled (and the tick labels are very small). This does not help in explaining the selection of $\beta$ (see next point).
    - Figure 7: Since the estimates are statistically reliable only for the first 800 action indices, a suggestion is to restrict the plot to this range so that it can be clearer.
1. The selection of threshold $\beta$ is described only in Appendix A.2. Given that $\beta$ is needed in both Section 3 (provenance tracing) and 4 (data selection), and given the claim (in bold) about the selection of $\beta$ (lines 442-445), this description should be in the main paper.
    - As a secondary point, the selection of threshold $\alpha$ is also unclear, specifically what is meant by "manual binary classification" (lines 243-244). What is the classification task? Why not set $\alpha$ larger than 0.1 if "the average probability difference for Shared
Sentence was found to be 0.097"?
1. Main results for data selection: With 3 teacher models x 4 student models x 2 data sources, there are 24 combinations. However, the main Table 1 shows only 4 combinations, and I did not find more in the appendix. There is thus no assurance that the combinations in Table 1 were not cherry-picked.
1. Some sentences are unclear or should otherwise be improved. Below I list only ones where I am quite unsure of what is meant. That said, I consider this point to be less important than the other presentation issues above.
    - Lines 155-156, "minimal sentence level": Why is it "minimal"? It is also not clear at this point that each reasoning step corresponds to a sentence.
    - Lines 253-254, "the Teacher Sentence curve in the figure": Which figure?
    - Lines 267-268, "Figure 2 and Figure 10 further illustrate this point": I do not see how these figures show emergence of other actions after a few steps.
    - Lines 320-327: This paragraph about LIMO-v2 is unclear to me, especially "LIMO-v2 contains only 800 samples." It appears that LIMO-v2 refers to both a model as well as a dataset. I do not understand the statements about the work being limited to models of $<8$B parameters. A quick look at LIMO-v2's GitHub page suggests the LIMO-v2 model has 32B parameters.
    - Lines 360-361, "for each question with multiple responses from teacher models": This is confusing because it suggests that multiple teachers give responses to the same question, and the selection is between teachers, which does not seem to be the case later on.
    - Section 4.2.2 ablation (3): What is the domain of the two main datasets, AceReason-1.1-SFT and OpenThought3-1.2M, if science is considered a different domain?

**Questions:**

In order of importance:
- Regarding weakness 4 above, how were the 4 combinations in Table 1 chosen?
- I would appreciate responses to the clarity issues identified above.
- Did the authors try splitting sentences using a natural language processor such as NLTK or spaCy? These may be more robust than the regex pattern used.

---

> ### Author Response · Authors · 2025-11-21
> **Responses to the Reviewer E2xa (Part 1 / 4)**
>
> > **Weakness 1**: The paper claims that distilled models reproduce teacher-originated actions in "entirely new contexts," and the newness of test contexts is emphasized throughout. However, this newness is not substantiated. Moreover, newness as a binary notion misses the fact that similarity to training contexts lies on a spectrum. I think it would be better to quantify somehow the dissimilarity of test contexts from training contexts, at both the sample and dataset levels, and analyze results in terms of this dissimilarity.
>
> We thank the reviewer for this helpful comment and apologize for the earlier ambiguity. We agree that our original wording around "entirely new contexts" may have been unclear due to differing interpretations, and we have revised the relevant text accordingly.
>
> To quantify the dissimilarity of test contexts from training contexts, we use LIMO‑v2 as an illustrative example. The training set of the LIMO-v2 model consists entirely of mathematical data, while its test set covers both mathematical reasoning and scientific reasoning. Therefore, we can **quantify cross-dataset relatedness by representing dataset text with TF–IDF vectors and computing cosine similarity to obtain the average similarity between datasets.**
>
> | |Similarity|
> |:-:|:-:|
> |AIME24 |0.0137|
> |GPQA-D|0.0039|
>
> For example, the similarity between the LIMO-v2 training set and the AIME24 test set is 0.0137, and that between the LIMO-v2 training set and the GPQA-D test set is 0.0039. Thus, **both AIME24 and GPQA-D have very low similarity to the training set, with GPQA-D, which we treat as an out-of-domain test set, having even lower similarity.** In this out-of-domain setting, the conclusions produced by our analysis framework remain consistent with those obtained in the in-domain setting.
>
>
> > **Weakness 2**: Figures are unclear, to the point of obscuring evidence for main claims of the paper (see especially the first point below).
>
> (1) We apologize for the issues with relevant Figures. **We have revised all these figures; please refer to the latest version of the PDF.**
>
> (2) Regarding the question "Can this correlation be quantified so that the evidence is not just a visual comparison of two curves?" we can compute the total difference in output probabilities for diffetent action types within the effective length range, which serves as a metric for quantifying the impact of the different action types.
>
> For example, in our analysis of DeepSeek-R1-0528-Qwen3-8B on AIME24, over the first 800 steps (those with relatively stronger statistical significance), we **compute the cumulative difference between the output probability of the all action types on correct versus incorrect answers and obtain the following sums:**
>
> ||Cumulative difference|
> |:-:|:-:|
> |Teacher Sentence|28.97|
> |Student Sentence|1.76|
> |Boosted Sentence|-31.64|
> |Shared Sentence|0.91|
>
> These results suggest that the **Teacher Sentence plays a stronger role in obtaining more correct answers.** Please refer to our response to Reviewer CXTo regarding Weakness 2 for additional applications of our method.
>
> (3) Regarding the comment on "understand finding (2) in Section 3.3": the yellow line consistently shows relatively high output probabilities across the four action types, where it is always ranked first or second. This indicates that the internal patterns of the student model are being reinforced. However, for the 7/8B model, the yellow line (Boosted Sentence) exhibits lower output probabilities on correct answers, whereas for the 32B model, the yellow line (Boosted Sentence) shows higher output probabilities on correct answers, which is consistent with the findings of the LIMO-v2 paper. In LIMO-v2 paper, the authors found that if a model already contains sufficient reasoning knowledge, reasoning distillation can activate it with minimal data scale. However, our results suggest that reinforcing certain internal patterns of the student model does not necessarily lead to performance improvements. The authors of LIMO-v2 paper also find that small models (e.g., 7B) trained with the same method in their paper fail to achieve good performance. In summary, **Student-internal patterns (Boosted Sentence) are often activated but not uniformly beneficial during reasoning distillation.**

---

> ### Author Response · Authors · 2025-11-21
> **Responses to the Reviewer E2xa (Part 2 / 4)**
>
> > **Weakness 3**: The selection of threshold is described only in Appendix A.2. Given that is needed in both Section 3 (provenance tracing) and 4 (data selection), and given the claim (in bold) about the selection of (lines 442-445), this description should be in the main paper.
>
>
> (1) Thank you for the suggestion. **We have incorporated the revised summary into the main body of the text (Lines 249-252)**, with detailed information provided in the appendix (spanning nearly two pages).
>
> (2) The parameter α is used to filter out relatively small probability differences that would otherwise affect the analysis. It is worth emphasizing that we only use this parameter during analysis. To obtain a relatively objective threshold, we asked 10 annotators to judge the magnitude of probability differences. For each annotation, the annotators were shown an output-probability line chart similar to that in Figure 2. Given all the sentences in the response, we asked them to identify those whose probability differences could be regarded as relatively negligible, such as sentences 55–65 in Figure 2. After we explained the definitions of the four sentence types and the goal of our analysis, the annotators manually selected the sentences that appeared more likely to be Shared Sentences, and we then derived the final (rounded) value of α from their judgments.
>
>
> > **Weakness 4**: Main results for data selection: With 3 teacher models x 4 student models x 2 data sources, there are 24 combinations. However, the main Table 1 shows only 4 combinations, and I did not find more in the appendix. There is thus no assurance that the combinations in Table 1 were not cherry-picked.
>
> We thank the reviewer for carefully reading our paper and apologize for the confusion caused by the insufficiently clear description of our experimental setting. To clarify, **we did not run all 24 possible combinations and then cherry-pick the best results.**
>
> **Choice of data sources and teacher models.** In practice, **each teacher model is tied to a specific dataset.** The training data we use mainly comes from two widely used open-source datasets: (i) NVIDIA's AceReason-1.1-SFT, which only provides responses generated by DeepSeek-R1, and (ii) OpenThought, which only provides responses generated by QwQ. Since teacher models are typically very large and produce long outputs, generating large-scale distillation data is computationally expensive. Therefore, we cannot afford to regenerate responses for these datasets using alternative teacher models.
> Beyond these two data settings, after the release of gpt-oss-120b, which has become one of the strongest open-source models in terms of benchmarks, we additionally sampled a subset of AceReason-1.1-SFT and generated responses with gpt-oss-120b to further evaluate our method under a state-of-the-art teacher model.
>
> **Choice of student models**. **(a)** Our early validation primarily used AceReason-1.1-SFT (with the dataset-provided DeepSeek-R1 responses) together with Qwen3-4B as the student model. After initially confirming the effectiveness of our method and completing the necessary ablation studies, we then conducted experiments on the larger Qwen3-8B model to further verify the generalization of our approach. **(b)** After completing the above experiments, in order to broaden both the dataset and teacher model coverage, we additionally adopted the OpenThought dataset (with the dataset-provided QwQ responses). We then naturally chose Qwen2.5-7B-Instruct as the student model, aligning with the setting in the OpenThought paper and using the same training hyperparameters to ensure the reliability of the training. **(c)** For AceReason-1.1-SFT with gpt-oss-120b as the teacher, we used Qwen3-4B-Instruct-2507 as the student model. The purpose is to combine a very strong teacher model with one of the strongest student models of similar size, so as to verify that our method remains effective even when the student model is already strong. **(d)** We did not train Qwen3-4B-Instruct-2507 on AceReason-1.1-SFT (with the dataset-provided DeepSeek-R1 responses) or on OpenThought (with the dataset-provided QwQ responses). As one of the latest models in the Qwen series, Qwen3-4B-Instruct-2507 already has strong reasoning ability, with performance close to or even surpassing that of DeepSeek-R1 and QwQ in some domains. Therefore, we believe that training Qwen3-4B-Instruct-2507 on data distilled from these earlier teacher models would have limited practical value for assessing our method.

---

> ### Author Response · Authors · 2025-11-21
> **Responses to the Reviewer E2xa (Part 3 / 4)**
>
> > **Weakness 5**: Some sentences are unclear or should otherwise be improved. Below I list only ones where I am quite unsure of what is meant. That said, I consider this point to be less important than the other presentation issues above.
>
> **We sincerely thank you for your detailed review and for your careful, conscientious evaluation. Below, we briefly address each of your comments in turn and will incorporate the corresponding revisions into the main manuscript.**
>
> *(1) Lines 155-156, "minimal sentence level": Why is it "minimal"? It is also not clear at this point that each reasoning step corresponds to a sentence.*
>
> We aim to analyze whether the output tokens of a distilled model match those of the teacher model. However, due to vocabulary differences between model families, we instead conduct the analysis at a slightly coarser granularity. In prior work (arXiv:2412.06559, [tutorial manual](https://help.aliyun.com/zh/pai/use-cases/distillation-solution-of-llm-for-deep-reasoning)) on reasoning models, segmentation is typically performed using "\n\n" as the delimiter. In our setting, this approach often produces segments containing many sentences, which complicates evaluation. Since our original aim was a token-level comparison, we instead design a segmentation scheme that produces relatively short segments so as to approximate token-level alignment, which motivates the term "minimal." Empirically, a segment usually coincides with a single sentence, with only a few cases in which it spans multiple sentences.
>
> *(2) Lines 253-254, "the Teacher Sentence curve in the figure": Which figure?*
>
> Here we provide an example that applies to all subfigures in Figures 3 and 4. In fact, this sentence can be omitted. We remove it.
>
> *(3) Lines 267-268, "Figure 2 and Figure 10 further illustrate this point": I do not see how these figures show emergence of other actions after a few steps.*
>
> In both figures, the first 10 sentences contain non–Teacher Sentence. This shows that although the insertion of <think> forces the first sentence to be a Teacher Sentence and may influence the subsequent sentences (since starting with <think> is a pattern the student model has never seen), this influence is limited. In Figure 2 and Figure 11 (previously Figure 10), The presence of other types of actions within these 10 sentences supports above claim.
>
> *(4) Lines 320-327: This paragraph about LIMO-v2 is unclear to me, especially "LIMO-v2 contains only 800 samples." It appears that LIMO-v2 refers to both a model as well as a dataset. I do not understand the statements about the work being limited to models of B parameters. A quick look at LIMO-v2's GitHub page suggests the LIMO-v2 model has 32B parameters.*
>
> Thank you for your suggestion. We will clearly distinguish between the LIMO-v2 model and the LIMO-v2 dataset.
>
> In addition, our experiments were conducted only on models with ≤8B parameters. Although we observed that Teacher Sentence also yields higher output probabilities on a 32B model, we did not provide quantitative evidence that 32B models would necessarily benefit from using more teacher sentences when training. Moreover, the LIMO-v2 dataset itself contains only 800 samples; a training set of this size can induce only modest changes in model behavior (indeed, even the differences between actions on correct vs. incorrect answers reported in the paper are relatively small) and is therefore insufficient to serve as quantitative evidence supporting our conclusions when training. Out of a sense of responsibility, we chose to make this limitation explicit.
>
> During the rebuttal phase, however, several reviewers raised this concern. As a supplementary yet computationally affordable study, we additionally sample 10k problems from OpenCodeReasoning, construct 10k training instances following our method, and train a larger model, Qwen3-30B-A3B-Base. Evaluated on the latest LiveCodeBench, the score improves from 41.98 (Vanilla) to 43.51 (Ours), providing further evidence for the effectiveness of our approach. For more details, please refer to our response to Reviewer 8GhT, Weakness 1.
>
> *(5) Lines 360-361, "for each question with multiple responses from teacher models": This is confusing because it suggests that multiple teachers give responses to the same question, and the selection is between teachers, which does not seem to be the case later on.*
>
> We apologize and have revised this sentence to ensure consistency with the subsequent description.
>
> *(6) Section 4.2.2 ablation (3): What is the domain of the two main datasets, AceReason-1.1-SFT and OpenThought3-1.2M, if science is considered a different domain?*
>
> We thank the reviewer for the reminder. These two datasets primarily consist of mathematical data, and we only use the questions from the math domain in these sources.

---

> ### Author Response · Authors · 2025-11-21
> **Responses to the Reviewer E2xa (Part 4 / 4)**
>
> > **Question 1**: Regarding weakness 4 above, how were the 4 combinations in Table 1 chosen?
>
> Please refer to our response to Weakness 4.
>
>
> > **Question 2**: I would appreciate responses to ；the clarity issues identified above.
>
> We greatly appreciate your diligence and effort, and we are very grateful for your review. Your comments have truly helped us improve the quality of the paper.
>
>
> > **Question 3**: Did the authors try splitting sentences using a natural language processor such as NLTK or spaCy? These may be more robust than the regex pattern used.
>
> Thank you very much for your suggestion. First, we do not explicitly pursue segmentation at the sentence level, because we believe that decoupling model outputs according to human semantic structures is not the only way to understand the model. Instead, our interpretation of the model outputs is grounded entirely in the model's own perspective. At the same time, we are aware that the teacher and student models may not always produce perfectly aligned token sequences, so we adopt a more relaxed strategy to address this issue. **We also provide a \[visualization\](https://anonymous.4open.science/r/AnonymizeRepository-DC7B/nltk_compare.png) of the distribution of the number of segments based on NLTK sentence segmentation**, which shows that **our segmentation method is broadly consistent with the results obtained by NLTK**. Once again, we appreciate your valuable suggestion. We believe that in many scenarios methods such as NLTK will be more robust, and we will add them as an optional sentence segmentation method.

---

> ### Author Response · Authors · 2025-11-28
>
> Dear Reviewer E2xa,
>
> We sincerely appreciate the time and effort you have devoted to reviewing our manuscript. Your insightful comments and suggestions have been extremely valuable to us, and we hope that our responses have been helpful in clarifying our work. As the discussion period draws to a close, we warmly welcome any further questions or comments you may have, and we would be very happy to provide additional clarification.
>
> Regarding the experimental settings, our intention, as a research-focused paper, is to use multiple settings to demonstrate the effectiveness of our method. In particular, most existing reports (e.g., HuggingFace OpenR1, a-m-team, NVIDIA AceReason, NVIDIA OpenMathReasoning, Alibaba OmniThought, Light-R1, LIMO, s1, Tencent DeepMath, MiroMind-M1, Syntheic-1, Meta NaturalThoughts, STILL-2, and Sky-T1) consider only a single setting (one teacher model with one student model, or several student models of different sizes). We believe that the consistent improvements observed across these different settings, together with our ablation studies, including the finding that increasing/decreasing Teacher Sentence yields the best/worst performance and the consistent gains across three different testset domains, provide evidence for the generalizability of our method.

---

### Official Review · Reviewer_GDUf · 2025-11-01

**Soundness:** 2
**Presentation:** 3
**Contribution:** 3
**Rating:** 4
**Confidence:** 4

**Summary:**

This paper investigates the origin of student model behaviors in reasoning distillation and introduces a framework called Reasoning Distillation Provenance Tracing (RDPT). Instead of only measuring performance, the authors analyze where each generated sentence (action) in a distilled model comes from—whether it originates from the teacher model, the student model, both, or is newly boosted through distillation. By comparing predictive probabilities from the teacher, student, and distilled models under the same context, they categorize actions into four provenance types. Using this analysis, the paper shows that distilled models indeed generate “teacher-originated” actions even in unseen contexts, and these actions are correlated with correctness, offering an interpretable explanation for distillation’s generalization benefits.
Building on this, the authors propose a teacher-guided data selection strategy that prioritizes examples with higher teacher–student divergence. Experiments across multiple teacher–student pairs (e.g., DeepSeek-R1, QwQ-32B, GPT-OSS-120B) and reasoning benchmarks (AIME24/25, GPQA-D, MATH500) demonstrate consistent performance improvements over prior heuristics such as GRAPE.

**Strengths:**

- Provides a novel analytical framework (RDPT) that quantitatively traces the provenance of reasoning actions, offering interpretability to the distillation process.
- Empirical evidence that distilled models reproduce teacher-originated behaviors in unseen contexts, explaining why reasoning distillation generalizes.
- The proposed teacher-guided data selection is simple yet principled, improving performance across diverse teacher–student configurations and datasets.
- Strong experimental design and evaluation, with systematic analysis on multiple benchmarks and detailed ablation studies.
- Offers insightful observations about how teacher- and student-originated patterns emerge and interact during reasoning, bridging the gap between empirical distillation and interpretability analysis.

**Weaknesses:**

- The proposed framework (RDPT) is primarily analytical and diagnostic rather than methodological. It provides interpretation of distillation outcomes but does not introduce new mechanisms that improve reasoning capability.
- The provenance classification relies on manually set thresholds (α, β) and simple probability gaps between teacher and student. This rule-based design is heuristic and potentially unstable across datasets or model scales.
- The definition of “teacher-originated” actions is superficial and token-level, lacking semantic or reasoning-structure grounding. High teacher probability does not necessarily imply genuine reasoning transfer.
- Experimental coverage is limited to ≤8B models and a narrow set of teacher–student pairs; it remains uncertain whether the findings generalize to stronger or cross-architecture settings.
- The teacher-guided data selection method is incremental over GRAPE, and its ~2% average gain is modest given the extra computational cost of obtaining teacher logits.
- The analysis focuses entirely on quantitative trends, without qualitative or behavioral validation to substantiate the claim that “teacher-originated” sentences truly reflect reasoning inheritance.
- The framework assumes access to teacher probabilities for all samples, which restricts scalability and practical applicability in closed-source or large-model scenarios.

**Questions:**

The experiments focus exclusively on mathematical reasoning tasks (AIME, MATH500, OlympiadBench, etc.), while reasoning distillation is also widely applied to other domains such as common-sense reasoning and natural language inference. Could the authors clarify why the analysis and validation were limited to math reasoning? Are there specific challenges that make extending the proposed provenance-tracing framework to broader reasoning domains difficult or less meaningful?

---

> ### Author Response · Authors · 2025-11-21
> **Responses to the Reviewer GDUf (Part 1 / 4)**
>
> > **Weakness 1**: The proposed framework (RDPT) is primarily analytical and diagnostic rather than methodological. It provides interpretation of distillation outcomes but does not introduce new mechanisms that improve reasoning capability.
>
> Thank you for your valuable comment.
>
> We would like to clarify that **our goal with RDPT is not only to analyze but also to directly guide and improve the distillation process**. In fact, **based on the insights from RDPT, we propose and validate a concrete method** that leads to measurable gains in distilled models’ reasoning performance. Concretely, RDPT identifies that a specific kind of output sentences (we termed as Teacher Sentence) has a substantial positive impact on the student’s reasoning ability. Subsequently, **we introduce a new method for selecting distillation data based on this insight**, and **validation experiments across different domains and teacher–student pairs consistently show performance improvements in Table 1 and Table 4 (for example, on AIME24, we achieve improvements of 2–4.9% across different settings.)**, demonstrating the effectiveness of this method in improving the reasoning capability of distilled models.
>
> In the context of distillation, the similarity between the student’s and teacher’s output probabilities naturally serves as an indicator of distillation quality and, to some extent, reflects how well the student model imitates the teacher’s behavior. Because the teacher has strong reasoning ability, this similarity will typically also indicate an improvement in the student model’s reasoning ability.
> **To further demonstrate the effectiveness of our new data selecting method, which is built upon RDPT,  in strengthening this similarity, we construct three sets of sentences**: the teacher model's outputs on the training set, the teacher model's outputs on the test set, and the distilled student model's outputs on the test set. For each set, we feed every response into both the teacher model and the distilled student model, **compute the difference between the output probabilities for each sentence**, and then **aggregate these differences to obtain the empirical distribution histogram of probability discrepancies. The results, shown in the figures, (\[Vanilla\](https://anonymous.4open.science/r/AnonymizeRepository-DC7B/random.jpg) and \[Ours\](https://anonymous.4open.science/r/AnonymizeRepository-DC7B/ours.jpg)), compare two different data-filtering strategies.**
>
> Under both strategies, the probability discrepancy is small on the training data (the teacher's outputs on the training set), which is expected because the training data are well fitted. The discrepancy on the teacher's outputs on the test set is slightly larger, indicating that, although the queries in the training and test sets differ, the student model still largely follows the teacher model's output distribution when evaluated on the teacher's context. Remarkably, **when evaluated on the distilled model's own context on the test set, our method achieves an even smaller probability discrepancy than on the training data. This strongly suggests a high degree of alignment with the teacher's output distribution.** It is important to note that the teacher distribution is not directly observable. We only approximate it indirectly via the probabilities assigned to sentences under different contexts. The fact, that the smallest probability discrepancy occurs on the distilled model's own outputs on the test set, indicates that our method allows the student model, in its own generation context, to closely align with the teacher model's output distribution.
>
> In addition, we believe that an analytical and diagnostic framework is also important in this area. In current reasoning distillation literature, most methods are largely heuristic: they propose recipes that empirically work, but provide little understanding of why certain traces or steps matter. This has led to a proliferation of ad-hoc techniques without a unifying view. RDPT contributes to this area by enabling a systematic evaluation of what constitutes better reasoning and better training data in the context of distillation, which most existing methods struggle to explain. Importantly, we show that **RDPT’s diagnostics are not merely descriptive: they correlate with and help predict downstream performance, and they directly inform the design of the improved distillation strategies**. Thus, **RDPT is both a conceptual framework and introduces a new mechanism that improve reasoning capability.**

---

> ### Author Response · Authors · 2025-11-21
> **Responses to the Reviewer GDUf (Part 2 / 4)**
>
> > **Weakness 2**: The provenance classification relies on manually set thresholds (α, β) and simple probability gaps between teacher and student. This rule-based design is heuristic and potentially unstable across datasets or model scales.
>
> Thank you for your valuable comment.
>
> **Regarding "relies on manually set" parameters**: we would like to clarify that every step of our approach is validated experimentally rather than based on ad-hoc manual choices.
>
> For the parameter α, its role is to prevent small probability differences from affecting the analysis results. It is worth emphasizing that we only use this parameter during analysis. To obtain a relatively objective value for α, we asked annotators to judge the magnitude of probability differences, and we derived α from their judgments. For the parameter β, in fact, **the threshold selection is fully automatic. In the revised version we have included a algorithm for this procedure.** Specifically:
>
> ```
> Input: teacher model, student model, and a batch of data.
> Output: optimal threshold.
>
> 1. Feed the data into the teacher model and the student model to obtain sentence-level output probabilities. Then, following Section 4.1, partition the sentences into Common Sentence, Teacher Sentence, and Student Sentence.
> 2. Initialize: best overlap area = ∞, best threshold = None.
> 3. For t in {0.05, 0.10, …, 1.0}:
>     - Compute the current overlap area = (overlap between Student Sentence and Common Sentence) +(overlap between Common Sentence and Teacher Sentence).
>     - If the current overlap area < best overlap area, set
> 	best overlap area = current overlap area, best threshold = t.
>     - if mean(Common Sentence) > mean(Teacher Sentence), break
> ```
>
> **We also additionally show the selection process for Table 1, Setting 4 in figure (https://anonymous.4open.science/r/AnonymizeRepository-DC7B/overlap.png).**
>
> The results in Table 3 and Figure 6 demonstrate that the threshold selected by our adaptive method indeed outperforms other thresholds. Please refer to Section 4.2.2 for more details.
>
>
> **Regarding "potentially unstable"**: using the same thresholding scheme, we perform analyses on four settings and experiments on five settings (with different training sets), where each experiment uses datasets of size 10k or 50k, which are not small in scale. All of these experiments achieved the desired results, indicating that the method is empirically stable and adapts well to different practical settings. Moreover, as shown in Table 3, even suboptimal choices of β still yield improved performance, highlighting the robustness of our method.
>
> > **Weakness 3**: The definition of "teacher-originated" actions is superficial and token-level, lacking semantic or reasoning-structure grounding. High teacher probability does not necessarily imply genuine reasoning transfer.
>
> Thank you for your valuable comment.
>
> **Regarding "token-level" and "lacking semantic or reasoning-structure grounding"**: We clarify that our method operates at the sentence level rather than the token level, and therefore naturally captures semantic information and reasoning structure. **We have provided an analysis from a semantic perspective where we observed that the Teacher Sentence, compared to other action types, exhibits relatively more Reflection behavior. More details in Appendix A.3.2, "What behaviors are included in different action types?".** While preserving semantic information, we also analyze distilled models through the perspective of model itself. Compared with purely semantic analyses, this provides a complementary and novel perspective. Therefore, our sentence categorization is not confined to a small set of predefined semantic behavioral categories, in contrast to prior work that relies on manually specified semantic rules for categorization. This provides a more flexible and adaptive way to analyze distilled models. For instance, "teacher-originated" sentences are not limited to phrases like "But/Wait, xxx," which are frequently emphasized in many papers.
>
> **Regarding reasoning transfer**: First, high teacher probability does not necessarily imply genuine reasoning transfer. Instead, in this paper, comparing the output probabilities across models provides a more intuitive, practical, and model-agnostic proxy for understanding the distilled model’s outputs.
>
> Second, we argue that the Teacher Sentence can indeed facilitate reasoning transfer. The similarity between the student’s and teacher’s output probabilities naturally reflects, to some extent, how well the student model imitates the teacher’s reasoning behavior. Higher similarity serves as indirect evidence of reasoning transfer and leads to corresponding performance gains. Our results further show that more Teacher Sentences can strengthen this similarity. Please refer to our response to Weakness 1 for more details.

---

> ### Author Response · Authors · 2025-11-21
> **Responses to the Reviewer GDUf (Part 3 / 4)**
>
> > **Weakness 4**: Experimental coverage is limited to ≤8B models and a narrow set of teacher–student pairs; it remains uncertain whether the findings generalize to stronger or cross-architecture settings.
>
> Thank you for your valuable comment. Our student models in the original version of paper are primarily restricted to ≤8B parameters, which we consider reasonable since the goal of distillation is precisely to enhance the reasoning ability of weaker, smaller student models. In practical usage, the most common setup is to distill a larger, stronger teacher model into a smaller student model, rather than using a student whose size is close to that of the teacher. Moreover, as noted in the paper, we did not conduct exhaustive experiments with larger-scale models due to limited computational resources.
>
> **As a supplementary yet computationally affordable study, we additionally sample 10k problems from OpenCodeReasoning, construct 10k training instances following our method, and train a larger model, Qwen3-30B-A3B-Base. Evaluated on the latest LiveCodeBench, the score improves from 41.98 (Vanilla) to 43.51 (Ours), providing further evidence for the effectiveness of our approach.**
>
> ||LiveCodeBench|
> |:-:|:-:|
> |Vanilla|41.98|
> |Ours|43.51|
>
> Our evaluation settings have encompassed a diverse set of models. (1) We experiment with both 4B and 8B variants of Qwen3, as well as the 30B model mentioned above, showing that our method generalizes across different parameter scales. (2) We conduct experiments on both Base and Instruct models, illustrating that our method is applicable to different model types. (3) Furthermore, we test models with varying initial capabilities (e.g., Qwen3-4B and Qwen3-4B-Instruct-2507), where Qwen3-4B-Instruct-2507 is among the strongest non-reasoning 4B student models, verifying that our method generalizes well across student models with different initial strengths. Regarding teacher models, we explore three distinct model families (Qwen, Deepseek, GPT-OSS), covering both dense and MoE architectures. Within the MoE architectures, we also examine both full-attention and sparse-attention variants, and observe that our method is effective across all of these settings.
>
> > **Weakness 5**: The teacher-guided data selection method is incremental over GRAPE, and its ~2% average gain is modest given the extra computational cost of obtaining teacher logits.
>
> Thank you for your valuable comment. Our teacher-guided data selection method is a direct and natural application of the conclusions drawn from our analysis framework. In a standard distillation pipeline, where responses are directly sampled from the teacher model, the corresponding output probabilities of the teacher model can be easily obtained. Our approach incurs computational costs similar to those of GRAPE. When must using existing data, our method incurs the additional cost of a single forward pass through the teacher model, which **is relatively small compared to the overall training costs**. For instance, in Table 1, Setting 3, we input all responses into the teacher model to obtain log probabilities (**\~45 GPU hours**) and into the student model to obtain log probabilities (**\~16 GPU hours**). Afterward, applying the method described in the paper, we select 50k examples, and training on these examples requires **approximately 600 GPU hours**.
>
> Furthermore, our method's design does not overlap with GRAPE in concrete terms. Motivationally, our approach is guided by a clearer optimization objective; a more detailed comparison can be found in Figure 5. Importantly, we observe improvements on almost all test sets, and an average gain of 2% on these difficult benchmarks is far from negligible. The score on MATH500 is already close to saturation, so the performance gain on this dataset is relatively small; we therefore suggest focusing on the other datasets for a more informative comparison. For instance, while Qwen3-32B represents a major upgrade over QwQ-32B (as part of a new version iteration in the Qwen series), its performance on AIME24 improves by only 1.9% (see Table 13 in the Qwen3 report). **GEAPE shows mixed performance on AIME24, with an average improvement of 0.65%. In contrast, our method consistently outperforms across all settings, with absolute gains ranging from 2% to 4.9%**.

---

> ### Author Response · Authors · 2025-11-21
> **Responses to the Reviewer GDUf (Part 4 / 4)**
>
> > **Weakness 6**: The analysis focuses entirely on quantitative trends, without qualitative or behavioral validation to substantiate the claim that "teacher-originated" sentences truly reflect reasoning inheritance.
>
> Thank you for your valuable comment.
>
> First we would like to clarify that **we have provided qualitative examples in Figures 2 and 11 (Figure 10 in the original version) and offered a semantic analysis of action types in Appendix A.3.2, titled "What behaviors are included in different action types?".**
>
> For example, in Figure 11, sentences belonging to the Teacher Sentence category are highlighted in green. We observe that sentences related to planning, reflection, and verification (e.g., "I need xxx", "I recall xxx", "But here, xxx", "Let me verify xxx") are more likely to fall into this category, and these behaviors are, in fact, relatively rare in the student model. By contrast, Shared Sentences primarily consist of execution behaviors. It is worth emphasizing that our analysis is based on model itself. Consequently, “teacher-originated” sentences are not confined to a small set of pre-defined semantic behavioral categories, which contrasts with prior work that relies on manually specified semantic rules for categorization. For instance, "teacher-originated" sentences are not limited to phrases like "But/Wait, xxx," which are frequently emphasized in many papers.
>
> > **Weakness 7**: The framework assumes access to teacher probabilities for all samples, which restricts scalability and practical applicability in closed-source or large-model scenarios.
>
> We acknowledge that access to the teacher's probabilities for its own generated samples is indeed required. However, **what we need is not the output probabilities over the entire vocabulary but rather the output probabilities for the target token alone.** For responses produced by the teacher model itself, **most closed-source systems (e.g., OpenAI, Gemini, Grok) already provide token-level probabilities directly during generation.** Moreover, the open-source large-model ecosystem is rapidly expanding: models such as DeepSeek-R1, Qwen3-235B, Kimi-1T, and GLM expose logits without restriction, offering a rich set of options for applying our method in practical scenarios.
>
> > **Question 1**: The experiments focus exclusively on mathematical reasoning tasks (AIME, MATH500, OlympiadBench, etc.), while reasoning distillation is also widely applied to other domains such as common-sense reasoning and natural language inference. Could the authors clarify why the analysis and validation were limited to math reasoning? Are there specific challenges that make extending the proposed provenance-tracing framework to broader reasoning domains difficult or less meaningful?
>
> We would like to clarify that **in the original version of our paper, we did not conduct analysis and experiments solely on the mathematics domain. We also analyzed and evaluated scientific reasoning** on GPQA-D (Figure 4 and Table 4), a PhD-level question‑answering dataset. **We further include evaluation on a code reasoning benchmark** (please refer to our response to Weakness 4) to emphasize that our method can be extended to a broader range of domains.
>
> Recent reasoning models are increasingly evaluated on highly challenging, competition-level benchmarks in math, science, and code, where multiple reasoning skills must be composed over long trajectories. In this setting, classic single-skill benchmarks such as common-sense reasoning or NLI often provide limited discrimination among strong models and are less representative of the practical targets of long cot reasoning distillation. For this reason, we primarily conduct our analysis and validation on math and science benchmarks, and we further complement them with experiments in the code domain. **Our benchmark setup is consistent with that of mainstream reasoning models, such as DeepSeek-R1, Qwen3, and OpenAI's o1.**
>
> We observe consistent improvements on challenging reasoning benchmarks across mathematics, code, and science, demonstrating that our method generalizes well to diverse reasoning domains.

---

> ### Author Response · Authors · 2025-11-28
>
> Dear Reviewer GDUf,
>
> We sincerely appreciate the time and effort you have devoted to reviewing our manuscript. Your insightful comments and suggestions have been extremely valuable to us, and we hope that our responses have been helpful in clarifying our work. As the discussion period draws to a close, we warmly welcome any further questions or comments you may have, and we would be very happy to provide additional clarification.
>
> In addition, we would like to clarify that the central focus of our paper is a detailed analysis of reasoning distillation itself, including both the new interpretability tool we propose (whether it is viewed as “primarily analytical and diagnostic” or “methodological”) and the subsequent improvements that follow from this analysis. Given that many recent industry reports (e.g., HuggingFace OpenR1, OpenThoughts, a-m-team, NVIDIA AceReason, NVIDIA OpenMathReasoning, Alibaba OmniThought, Light-R1, LIMO, s1, Tencent DeepMath, MiroMind-M1, Syntheic-1, Meta NaturalThoughts, STILL-2, and Sky-T1) are also concerned with this topic, we believe that both our new perspective and the resulting improvements on test domains aligned with those used in these reports are of independent value.

---

### Official Review · Reviewer_8GhT · 2025-11-01

**Soundness:** 4
**Presentation:** 3
**Contribution:** 4
**Rating:** 8
**Confidence:** 4

**Summary:**

This paper introduces Reasoning Distillation Provenance Tracing, a novel analytical framework for understanding where the behaviors of a distilled reasoning model originate, whether from the teacher model, the student’s prior patterns, or both. Building on these findings, they propose a teacher-guided data selection strategy, experiments across multiple teacher–student pairs show consistent improvements.

**Strengths:**

- The paper is well organized and good written
 - RDPT introduces an interpretable framework for tracing the origin of reasoning steps, bridges the gap between explainability and distillation efficiency.
 - Proposes a simple yet effective teacher-guided data selection method.

**Weaknesses:**

- Lack of validation on larger-scale (e.g., 70B+) models questions scalability
 - The proposed teacher-guided selection requires re-feeding large corpora through both models for probability extraction, cost analysis is needed
 - The method is primarily validated on short- to medium-length reasoning traces, leaving its effectiveness in long-range or multi-step reasoning scenarios uncertain.

**Questions:**

Does producing more Teacher Sentences **cause** better reasoning, or merely co-occurs with it?

---

> ### Author Response · Authors · 2025-11-21
> **Responses to the Reviewer 8GhT (Part 1 / 2)**
>
> > **Weakness 1**: Lack of validation on larger-scale (e.g., 70B+) models questions scalability
>
> Thank you for your valuable comment. Our student models in the original version of paper are primarily restricted to ≤8B parameters but we believe this is a reasonable approach, as the goal of distillation is specifically to enhance the reasoning ability of smaller, weaker student models. In practical applications, the most common setup is to distill a larger, more powerful teacher model into a smaller student model, rather than using a student model that is similar in size to the teacher. Additionally, we did not conduct exhaustive experiments on larger-scale models due to limited computational resources.
>
> **As a supplementary yet computationally affordable study, we additionally sample 10k problems from OpenCodeReasoning, construct 10k training instances following our method, and train a larger model, Qwen3-30B-A3B-Base. Evaluated on the latest LiveCodeBench, the score improves from 41.98 (Vanilla) to 43.51 (Ours), providing further evidence for the effectiveness of our approach.**
>
> ||LiveCodeBench|
> |:-:|:-:|
> |Vanilla|41.98|
> |Ours|43.51|
>
>
> Our evaluation settings have encompassed a diverse set of models. (1) We experiment with both 4B and 8B variants of Qwen3, as well as the 30B model mentioned above, showing that our method generalizes across different parameter scales. (2) We conduct experiments on both Base and Instruct models, illustrating that our method is applicable to different model types. (3) Furthermore, we test models with varying initial capabilities (e.g., Qwen3-4B and Qwen3-4B-Instruct-2507), where Qwen3-4B-Instruct-2507 is among the strongest non-reasoning 4B student models, verifying that our method generalizes well across student models with different initial strengths. Regarding teacher models, we explore three distinct model families (Qwen, Deepseek, GPT-OSS), covering both dense and MoE architectures. Within the MoE architectures, we also examine both full-attention and sparse-attention variants, and observe that our method is effective across all of these settings.
>
> > **Weakness 2**: The proposed teacher-guided selection requires re-feeding large corpora through both models for probability extraction, cost analysis is needed
>
> We appreciate the reviewer's suggestion and will include a cost analysis in the final version. In a standard distillation pipeline, where we should sample responses directly from the teacher model, we can directly obtain the corresponding output probabilities of teacher model. Our approach only requires re-feeding the data into the student model. When must using existing data, the computation time depends on the size of the teacher model, the response length, and the number of GPUs used. Nevertheless, in all cases, **the cost of these forward passes is substantially lower than the overall training cost.** For instance, in Table 1, Setting 3, we feed all responses into the teacher model to obtain log probabilities (**\~45 GPU hours**) and into the student model to obtain log probabilities (**\~16 GPU hours**). We then apply the method described in the paper to select 50k examples, which are subsequently used for training (**\~600 GPU hours**).
>
>
>
> > **Weakness 3**: The method is primarily validated on short- to medium-length reasoning traces, leaving its effectiveness in long-range or multi-step reasoning scenarios uncertain.
>
> Thank you for your valuable comment. Our training data is generated by state-of-the-art long CoT reasoning models, with their reasoning traces can be as long as **64k tokens (as shown in Table 5) or thousands of steps**, which even exceeds the maximum context length of a large part of LLMs. Moreover, our evaluation sets are extremely challenging, including mathematical olympiad problems (AIME) and PhD-level scientific reasoning QA (GPQA), all of which require complex, multi-step reasoning. We will emphasize this point more clearly in the revised version to avoid potential misunderstandings.

---

> ### Author Response · Authors · 2025-11-21
> **Responses to the Reviewer 8GhT (Part 2 / 2)**
>
> > **Question 1**: Does producing more Teacher Sentences cause better reasoning, or merely co-occurs with it?
>
>
> Thank you for this valuable comment.
>
> First, **Table 2 in our paper already provides evidence that Teacher Sentences do not merely co-occur with better reasoning, but that an increased number of Teacher Sentences is strongly associated with better reasoning.** In Table 2, we consider three configurations with the largest, smallest, and an intermediate number of Teacher Sentences, respectively, and we observe that the performance changes are consistent with the changes in the number of Teacher Sentences. This suggests that the two are not simply co-occurring phenomena (e.g., it is not the case that better performance happens to coincide with more Teacher Sentences by chance).
>
> **In the distillation setting, we view better reasoning as having two aspects: (i) the student model should learn the teacher model’s output distribution, and (ii) this alignment should translate into better performance on the test set.** While Table 2 already shows that Teacher Sentences are associated with better test performance (aspect (ii)), we **additionally conduct experiments targeting aspect (i)** to further clarify the relationship between Teacher Sentences and better reasoning.
>
> Specifically, to examine aspect (i), **we construct three sets of sentences**: the teacher model's outputs on the training set, the teacher model's outputs on the test set, and the distilled student model's outputs on the test set. For each set, we feed every response into both the teacher model and the distilled student model, **compute the difference between the output probabilities for each sentence**, and then **aggregate these differences to obtain the empirical distribution histogram of probability discrepancies.**
>
> **The results, shown in the figures, (\[Vanilla\](https://anonymous.4open.science/r/AnonymizeRepository-DC7B/random.jpg) and \[Ours\](https://anonymous.4open.science/r/AnonymizeRepository-DC7B/ours.jpg)), compare two different data-filtering strategies.** Under both strategies, the probability discrepancy is small on the training data (the teacher's outputs on the training set), which is expected because the training data are well fitted. The discrepancy on the teacher's outputs on the test set is slightly larger, indicating that, although the queries in the training and test sets differ, the student model still largely follows the teacher model's output distribution when evaluated on the teacher's context. Remarkably, **when evaluated on the distilled model's own context on the test set, our method achieves an even smaller probability discrepancy than on the training data. This strongly suggests a high degree of alignment with the teacher's output distribution.** It is important to note that the teacher distribution is not directly observable. We only approximate it indirectly via the probabilities assigned to sentences under different contexts. The fact, that the smallest probability discrepancy occurs on the distilled model's own outputs on the test set, indicates that our method allows the student model, in its own generation context, to closely align with the teacher model's output distribution. **In summary, compared to the absence of more Teacher Sentences, the inclusion of more Teacher Sentences leads to (1) improved performance on the test set, and (2) better similarity in output probabilities.**

---

> ### Author Response · Authors · 2025-11-28
>
> Dear Reviewer 8GhT,
>
> We sincerely appreciate the time and effort you have devoted to reviewing our manuscript. Your insightful comments and suggestions have been extremely valuable to us, and we hope that our responses have been helpful in clarifying our work. As the discussion period draws to a close, we warmly welcome any further questions or comments you may have, and we would be very happy to provide additional clarification.
>
>
> Since you have expressed interest in whether the relationship between the Teacher Sentence and the correctness of the final answer is merely correlational or actually causal, we would like to emphasize our current conclusion. In the paper, we deliberately state our findings in a conservative way, claiming only that there is a correlation between the presence of the Teacher Sentence and improved performance. However, in controlled experiments, we observe that increasing the proportion of Teacher Sentences consistently leads to better performance, while decreasing their proportion leads to worse performance. This suggests that the Teacher Sentence is likely to be one of the important contributors to the performance gains, which is also intuitively reasonable: learning more from the teacher model can lead to better performance. That said, we do not claim to have fully established strict causal identification at this stage, and we plan to conduct more fine-grained controlled experiments in future work.

---

### Official Review · Reviewer_CXTo · 2025-11-06

**Soundness:** 3
**Presentation:** 3
**Contribution:** 3
**Rating:** 6
**Confidence:** 2

**Summary:**

This paper considers the problem of tracing provenance in LLM reasoning distillation. The paper proposes Reasoning Distillation Provenance Tracing, a method that quantifies the sources of a distilled model's capabilities: whether it is from the teacher, from the student, or both. Based on this, the paper then proposes a new teacher-guided data selection method, showing its efficacy across various teacher/student models.

**Strengths:**

- A timely, yet interesting topic
- Well written
- Several qualitatively interesting experimental results, and promising results for the newly proposed data selection method

**Weaknesses:**

- Although I'm not an expert in this field, it is still clear that the paper is lacking discussions of prior literature on knowledge distillation and model auditing/provenance [1,2]. Additionally, core literature on distilling LLM reasoning capabilities [3,4] is lacking.
- Can the proposed provenance methodology be used to identify which teacher model distilled the reasoning capability?


[1] https://dl.acm.org/doi/10.1145/3292500.3330885

[2] https://openreview.net/forum?id=TatRHT_1cK

[3] https://aclanthology.org/2023.acl-long.830/

[4] https://aclanthology.org/2023.findings-acl.507/

**Questions:**

1. All the provenance analyses are done with open-source, already-distilled models only. What are the effects of different distillation methods? Do they lead to different provenance?
2. If one utilizes distillation methodologies that explicitly matches sequential-level probabilities [5,6], then would this lead to more homogeneous provenance and potentially even better data selection? Some discussions on this and references therein would be helpful as well.



[5] https://proceedings.mlr.press/v235/ko24c.html

[6] https://proceedings.mlr.press/v267/ko25a.html

---

> ### Author Response · Authors · 2025-11-21
> **Responses to the Reviewer CXTo (Part 1 / 3)**
>
> > **Weakness 1**: Although I'm not an expert in this field, it is still clear that the paper is lacking discussions of prior literature on knowledge distillation and model auditing/provenance [1,2]. Additionally, core literature on distilling LLM reasoning capabilities [3,4] is lacking.
>
> We appreciate the valuable suggestions. **In the latest version of the PDF, we have revised the previous Related Work and added more detailed discussions** on knowledge distillation, model auditing, and core literature on distilling LLM reasoning capabilities.
>
> In papers [1, 2], the primary focus is on analyzing the outputs of a single model and performing auditing/provenance to trace the source of training data. In contrast, our method conducts a cross-model analysis of outputs from multiple models, focusing on tracing the model-level source of outputs within a distillation setting. Papers [3, 4] primarily focus on improving distillation of LLM reasoning capabilities by designing more explicit chain-of-thought structures. In our work, we adopt the common practice of using the predefined "\<think\>thinking content\<\/think\>answer content" format for current reasoning models.
>
> **We have also incorporated these comparisons into the Related Work section. For further details, please refer to the latest version of the PDF.**
>
> > **Weakness 2**: Can the proposed provenance methodology be used to identify which teacher model distilled the reasoning capability?
>
> Thank you for this interesting question. **In short, our method is in principle capable of providing evidence about which teacher model contributed to a given reasoning capability.** To demonstrate this ability, **we consider two different scenarios** and **validate it in each of them.**
>
> (1) The student model is given, but its teacher model are unknown.
>
> We employ the distilled model from setting 1 in Table 1, which was in fact distilled from DeepSeek-R1. We then apply our analysis method to two candidate teacher models (i.e., DeepSeek-R1 and gpt-oss-120b) to determine which teacher was actually used for distillation. **The analysis results for DeepSeek-R1 are shown in the figure at https://anonymous.4open.science/r/AnonymizeRepository-DC7B/ture_R1.png, while results for gpt-oss-120b are presented in the figure at https://anonymous.4open.science/r/AnonymizeRepository-DC7B/fake_gptoss120b.png**. We observe that, when a spurious teacher model is used, the output probability of the Teacher Sentence is the lowest, while the output probability of the Student Sentence is relatively high. **This indicates that our analysis method can, to some extent, identify the true teacher model behind a distilled student model.**
>
> (2) The student model is known to be distilled from multiple teacher models.
>
> From a methodological perspective, our provenance analysis can partially identify the extent to which the teacher model contributes to the student model's reasoning capability. For example, in our analysis of the LIMO-v2 model, we observe that, across all teacher models, the Teacher Sentence tends to be assigned a higher output probability on correct answers than on incorrect ones. However, different teacher models exert heterogeneous influences on the student model. Concretely, over the first 400 steps (those with relatively stronger statistical significance), **we compute the cumulative difference between the output probability of the Teacher Sentence on correct versus incorrect answers, and obtain the following sums:**
>
> |  | Cumulative difference |
> |:-|:-:|
> |gpqa-R1 | 9.41|
> |gpqa-QwQ | 7.66|
> |aime-R1 | 7.09|
> |aime-QwQ | 7.88|
>
> These results suggest that, although the exact proportions of R1- and QwQ-generated data in the LIMO-v2 training set are unknown, the data produced by these two teacher models have heterogeneous effects across domains. We tentatively hypothesize that, **on GPQA, the reasoning capability distilled from R1 is particularly beneficial for scientific questions.** The result is consistent with the fact that DeepSeek-R1 substantially outperforms QwQ on GPQA while performing comparably to QwQ on AIME. In future work, we plan to further investigate how to identify which teacher model has contributed the distilled reasoning capability to the student model on different domains.

---

> ### Author Response · Authors · 2025-11-21
> **Responses to the Reviewer CXTo (Part 2 / 3)**
>
> > **Question 1**: All the provenance analyses are done with open-source, already-distilled models only. What are the effects of different distillation methods? Do they lead to different provenance?
>
> Thank you for your valuable comment.
>
>
> **Our analyse method is distillation-method agnostic**. First we would like to clarify that our provenance framework is inherently distillation-method agnostic: it relies only on the student model's static states before and after distillation, together with the teacher model's output probability and does not depend on the specific distillation method used to train the student model.
>
>
> Current analysis is primarily based on the most commonly used distillation method for LLMs. In the original submission, our main analyses are conducted on open-source distilled models, specifically DeepSeek-Distill-Qwen-7B, DeepSeek-R1-0528-Qwen3-8B and LIMO-v2 models. All of the models are trained using the most commonly used distillation method for LLMs (SFT on teacher-generated data). DeepSeek pioneered reasoning distillation by demonstrating that supervised fine-tuning on teacher model outputs can dramatically enhance the reasoning abilities of smaller models. Numerous projects (e.g., HuggingFace OpenR1, OpenThoughts, a-m-team, NVIDIA AceReason, NVIDIA OpenMathReasoning, Alibaba OmniThought, Light-R1, LIMO, s1, Tencent DeepMath, MiroMind-M1, Syntheic-1, Meta NaturalThoughts, STILL-2, and Sky-T1) have since devoted substantial effort to constructing and refining large-scale corpora of challenging reasoning questions paired with teacher responses, using rigorous quality filtering, correctness checks, and diversity-conscious curation. This paradigm of training on carefully curated teacher-generated data has produced distilled models that achieve state-of-the-art or highly competitive performance across diverse domains while retaining the efficiency of compact model architectures. Therefore, in this paper we primarily focus on this paradigm.
>
> Another paradigm is logits distillation, a classical approach in knowledge distillation that aligns the logit distribution of student and teacher model. However, this approach faces practical challenges on LLMs when the teacher and student employ different tokenizers or vocabularies, as direct logit alignment becomes infeasible due to misaligned output spaces, which limits its use. In addition, this method typically requires storing the teacher's logits over the entire vocabulary, creating substantial engineering overhead and making it hard to deploy at scale, especially for state-of-the-art reasoning LLMs. Since we focus on a broader notion of distillation, allowing arbitrary teacher-student pairs across tokenizers and architectures, we did not conduct a dedicated analysis of this particular distillation approach. Whether different distillation methods lead to different provenance patterns is indeed an interesting question. To investigate this, **we additionally analyze Qwen3-4B**, which, as stated in the Qwen3 technical report, is distilled from Qwen3-32B using a variant of logits distillation. **The results are shown in the figure at https://anonymous.4open.science/r/AnonymizeRepository-DC7B/Qwen-4B-Base----logits_method.png**. Overall, **the conclusions we obtain from this logits distilled model are consistent with those reported in the Section 3.3.**

---

> ### Author Response · Authors · 2025-11-21
> **Responses to the Reviewer CXTo (Part 3 / 3)**
>
> > **Question 2**: If one utilizes distillation methodologies that explicitly matches sequential-level probabilities [5,6], then would this lead to more homogeneous provenance and potentially even better data selection? Some discussions on this and references therein would be helpful as well.
>
> Thank you for your valuable comment. **[5, 6] are variants of the classical logits distillation, and we have added a discussion of them in the related work section.** However, as we explained in our response to Question 1, these methods face substantial limitations when applied to industrial-scale reasoning LLMs. Consequently, we did not include a direct empirical analysis of these approaches.
>
> (1) As an alternative, **we have added analysis for Qwen3-4B**, an open-source model distilled using a similar logits distillation methodology. **The analysis results is available at https://anonymous.4open.science/r/AnonymizeRepository-DC7B/Qwen-4B-Base----logits_method.png**. **Setting 1 in Table 1 can serve as a comparison**, which is obtained by performing SFT on the same student model using teacher-generated responses. **The corresponding analysis is available at https://anonymous.4open.science/r/AnonymizeRepository-DC7B/Qwen-4B-Base----sft_method.png.**
>
> Based on the analysis, **logits distillation has the potential to lead to more homogeneous provenance.** The evidence is that the Teacher Sentence exhibits a larger difference in output probability between correct and incorrect answers. However, we would like to emphasize that **this should not be viewed as a rigorous conclusion, since training performance is influenced by many factors, such as data scale, hyperparameters, and other implementation details**. In any case, this is an interesting question and we plan to investigate it more thoroughly in future work.
>
> (2) Implications for data selection. Regarding "better data selection," because different model families use different vocabularies, and these vocabularies are very large, the storage cost of retaining full vocabularies is high. As a result, directly combining sequential-level probability matching methods is challenging. Nevertheless, we believe this provides a new perspective, and there may exist approximate approaches, such as storing probabilities only for the top 20 tokens. We once again appreciate your suggestion and plan to explore this direction in more depth.

---

> ### Comment · Reviewer_CXTo · 2025-11-27
>
> Thank you for the detailed responses and for the additional experiments, which have addressed all of my concerns. After reading through the responses and other reviewers' comments, I am raising my score from 6 to 8. Since I'm not an expert (especially with the specifics of the LLM experiments), I keep my confidence as it is.
>
> One other question: the authors mention that setting $\alpha$ involves manual human annotations. Does this have to be done for every new experimental setup?

---

> ### Author Response · Authors · 2025-11-28
>
> Dear Reviewer CXTo:
>
> Thank you for your valuable feedback and for taking the time to review our updated work. We greatly appreciate that you raised your score. Thank you again for your valuable comments and encouragement. We sincerely appreciate your support.
>
> The parameter α is used to filter out relatively small differences in probabilities that would otherwise affect the analysis, and it is applied only during the analysis stage.  We did not re-annotate the data for each new experimental setting because we found that α = 0.1 is an appropriate value for filtering relatively small probability differences.  We fix α = 0.1 for all settings;  this choice does not affect the main conclusions of the paper across different settings.

---

### Meta-Review · Area_Chair_m2ix · 2026-01-07

**Summary:**

It is extremely complicate to summarize concerns on this paper, as reviewers have many different issues.

**Reviewer Concerns:**

Focussing on less positive reviewers:

R GDUf: not a methodological paper, that is, there is not a proposal -  **Authors answered to this concern**

R GDUf: lack of semantic/reasoning grounding - **in my opinion, the authors' response is generic**

R GDUf: Not qualitative analysis -  **Authors answered to this concern**

R E2xa: Many stylistic issues -  **Authors answered to this concern**

**Reviewer Scores:**

If R GDUf had participated in the discussion, they could have increased the score

If R E2xa had participated in the discussion, they could have increased the score

---

### Decision · Program_Chairs · 2026-01-26

Accept (Poster)